# Source characterization of volatile organic compounds measured by PTR-ToF-MS in Delhi, India

Liwei Wang[1], Jay G. Slowik[1], Nidhi Tripathi[2], Deepika Bhattu[1], Pragati Rai[1], Varun Kumar[1], Pawan Vats[4], Rangu Satish[5,a], Urs Baltensperger[1], Dilip Ganguly[4], Neeraj Rastogi[5], Lokesh K. Sahu[2], Sachchida N. Tripathi[3,*], and André S. H. Prévôt[1,*]

[1] Laboratory of Atmospheric Chemistry, Paul Scherrer Institute, Villigen, 5232, Switzerland

[2] Space and Atmospheric Sciences Division, Physical Research Laboratory, Ahmedabad 380009, India

[3] Department of Civil Engineering and Centre for Environmental Science and Engineering, Indian Institute of Technology Kanpur, Kanpur 208016, India

[4] Centre for Atmospheric Sciences, Indian Institute of Technology Delhi, New Delhi 110016, India

[5] Geosciences Division, Physical Research Laboratory, Ahmedabad 380009, India

[a] Now at: Department of Environmental Science and Analytical Chemistry, Stockholm University, Stockholm 11419, Sweden

[*] Corresponding authors

**Abstract.** Characteristics and sources of volatile organic compounds (VOCs) were investigated with highly
time-resolved simultaneous measurements by two proton-transfer-reaction time-of-flight mass spectrometers
(PTR-ToF-MS) at an urban and a suburban site in New Delhi, India from January to March 2018. During the
measurement period, high mixing ratios of VOCs and trace gases were observed, with high nocturnal mixing
ratios and strong day-night variations. The positive matrix factorization (PMF) receptor model was applied
separately to the two sites, and six major factors of VOCs were identified at both sites, i.e., two factors related to
traffic emissions, two to solid fuel combustion, and two secondary factors. At the urban site, traffic-related
emissions comprising mostly mono-aromatic compounds were the dominant sources, contributing 56.6% of the
total mixing ratio, compared to 36.0% at the suburban site. Emissions from various solid fuel combustion
processes, particularly in the night, were identified as a significant source of aromatics, phenols and furans at
both sites. The secondary factors accounted for 15.9% of the total VOC concentration at the urban site and for
33.6% at the suburban site. They were dominated by oxygenated VOCs and exhibited substantially higher
contributions during daytime.



**Key words:** VOCs, Source apportionment, traffic emission, solid fuel combustion, secondary formation, Delhi

## 1 Introduction

Volatile organic compounds (VOCs) are important trace gas constituents in the troposphere, impacting local and regional air quality, human health and climate both directly and indirectly (IPCC, 2013). With the participation of $NO_x$, oxidation of VOCs leads to the formation of tropospheric $O_3$, causing regional photochemical smog (Atkinson, 2000; de Gouw et al., 2005). The chemical transformation of VOCs forms less-volatile compounds and can contribute to gas-to-particle partitioning either by new particle formation or condensation on existing particles (Hallquist et al., 2009; Ehn et al., 2014). In addition, many VOCs are toxic, such as aromatic compounds, and exposure to large amounts of VOCs may adversely affect human health, including acute and chronic effects on different systems and organs, and even cancer (Kampa and Castanas, 2008; Nurmatov et al., 2013).

VOCs originate from both natural and anthropogenic sources. Biogenic VOCs (BVOCs), mainly emitted by plants, are regarded as the largest source of VOCs globally (Atkinson and Arey, 2003; Hallquist et al., 2009). However, in urban areas, anthropogenic VOCs can be dominant (Borbon et al., 2013). Vehicular exhaust emissions have long been regarded as the dominant source of VOCs in many urban areas. Many of these compounds are reactive and thus contribute significantly to urban $O_3$ pollution, photochemical smog, and secondary organic aerosol (SOA) formation (Fraser et al., 1998; Derwent et al., 2010; Müller et al., 2012). Biomass burning is considered as the second largest source of VOCs worldwide (Crutzen and Andreae, 1990) and could be a major VOC source in some urban areas during biomass burning events (Karl et al., 2007; Yokelson et al., 2009; Baudic et al., 2016; Languille et al., 2019). Compared to vehicular emissions, biomass burning sources emit more oxygenated and high molecular-weight VOCs such as furans and phenols. Due to high atmospheric reactivity and higher SOA yield, these compounds can also contribute substantially to SOA formation (Sekimoto et al., 2018; Koss et al., 2018).

Air pollution in South Asia has attracted more and more attention in recent years. This region is regarded as one of the most polluted regions in the world (e.g., Monks et al., 2009). Due to rapid urbanization and the lack of widespread advanced pollution control technologies in the industrial, energy, and transportation sectors (Mahata et al., 2018), air pollution has become an increasingly important issue across the region, particularly in large urban areas. The increasing emissions of air pollutants not only impact the local/regional air quality and human health, but also influences distant and pristine areas through transport (Bonasoni et al., 2010; Lawrence and Lelieveld, 2010; Mahata et al., 2018). Delhi, the capital city of India, with a population of 18.98 million (2012) people, is facing an air quality problem ranked as the worst among 1600 major cities in the world (WHO, 2014). The critical air quality problems have left India with high death rates from chronic obstructive pulmonary disease (COPD) and respiratory disease e.g., asthma (WHO, 2014). Despite the severe air pollution in Delhi, information on pollution levels, as well as an emission inventory of VOCs, and their sources in Delhi is still lacking (Monks et al., 2009; Kumar et al., 2015). A few studies on BTEX (i.e. benzene, toluene, ethylbenzene, and xylene) in Delhi reported that vehicular emissions could be the dominant source in Delhi (Srivastava et al., 2005; Hoque et al., 2008). Several studies also identified motor vehicle emissions as a significant source of particulate matter (PM), with high contributions from solid fuel combustion and industrial emissions (Sahu et al., 2011; Sharma et al., 2014; Kumar et al., 2015). These studies reveal the importance of local anthropogenic sources. However, since previous studies mainly focused on certain families of VOCs or a few VOCs species, a

comprehensive investigation of the VOC pollution levels, specific emission sources, as well as their roles in the
local tropospheric chemistry has not been reported previously. VOC source apportionment studies have been
conducted only in a few Indian cities. For example, Srivastava studied 23 VOC compounds in Mumbai city, and
pointed out the lack of evaporative and oceanic emissions (Srivastava, 2004). Sahu and Saxena (2015) studied
15 VOC species measured by a PTR-ToF-MS in urban Ahmedabad in winter. A recent study on 32 VOCs at
Mohali (a suburban site in northwest Indo-Gangetic Plain) identified biofuel usage, biomass burning and
vehicular emissions as important primary sources, and pointed out the lacking of present emission inventories
(Pallavi et al., 2019).
In this study, we report simultaneous on-line measurements of VOCs using two PTR-ToF-MS instruments at an
urban and a suburban site of Delhi, India. The level, composition and source characteristics of different VOCs
(158 ions at IITD and 90 ions at MRIU) were analyzed with the aid of the positive matrix factorization (PMF)
model. Spatial and temporal comparisons between the two sites are discussed for both the sources and selected
ions.
**2 Experiments and methods**
**2.1 Measurement sites**
Figure 1 shows the studied region and locations of the two measurement sites. The measurements were
conducted from 18 January to 10 March 2018 at an urban site at the Indian Institute of Technology (IITD), New
Delhi, and from 16 January to 8 March 2018 at a suburban site at Manav Rachna International University
(MRIU), Faridabad (Fig. 1).
At the urban site (28° 33'N, 77° 12'E), the inlet system was installed on the rooftop (~20 m above the ground) of
block VI (a four-story building), Centre for Atmospheric Sciences, on the campus of IITD (Gani et al., 2019;
Rai et al., 2020). The IITD campus is located in the southern part of the city center, and is surrounded by
educational, commercial, and residential districts. The study site is approximately 80 m north of a busy street
and is surrounded by several streets inside the campus as well. Thus, in addition to vehicular emissions,
commercial and residential activities may also produce VOC emissions in the immediate vicinity. The sampling
line of the PTR-ToF-MS was approximately 1.5 m long, and the inlet consisted of polyether ether ketone
(PEEK) tubing, with an inner diameter of 0.075 mm.
The other measurement site, at MRIU, was located in a relatively open area in suburban Delhi, about 20 km
southeast of the IITD site. Besides, the northeast territory of MRIU is of slightly higher elevation compared to
the sampling site, as shown in Fig. 1. The suburban site is located inside a big campus, and it is surrounded by
several small parks, with only a few narrow roads nearby. Although the site is not far from the main road, the
traffic load and other anthropogenic activities nearby may be much less than that at IITD. The instruments were
located on the first floor of a teaching building on the campus of MRIU. A similar PEEK inlet as at IITD was
used, with the sampling line approximately 2 m long.
**2.2 Online instruments**
Data collected in this study included the mixing ratios of VOCs, $NO_x$, CO, and meteorological parameters. The
involved monitors, analyzers, and sensors are described in detail in the following. Two PTR-ToF-MS 8000
(Ionicon Analytical G.m.b.H, Innsbruck, Austria) were simultaneously deployed at the two sites. At both sites,
the PTR-ToF-MS instruments were operated in the $H_3O^+$ mode, where the sampled VOCs are protonated via
non-dissociative proton transfer from $H_3O^+$ ions (Eq.1):
$$H_3O^+ + R \rightarrow RH^+ + H_2O \tag{1}$$
The PTR-ToF-MS measures non-methane organic gases (NMOG) with a proton affinity higher than water, i.e.,
most of the common VOCs such as carbonyls, acids, and aromatic hydrocarbons, as well as alkanes with more
than eight carbons and alkenes with more than two carbons. A detailed description of the instrument is found in
Jordan et al. (2009) and Graus et al. (2010). For both sites, the time resolution was set to 30 s, the drift tube
voltage was set at 600 V, with a drift temperature of 60 ℃, and a pressure of 2.2-2.3 mbar, resulting in a
reduced electric field (*E/N*) value of about 130 Td. Therefore, a similar fragmentation pattern is expected.
Calibrations were performed twice at the IITD site and three times at the MRIU site by dynamic dilution of
VOCs using a certified 15-compound gas standard (Ionimed Analytik GmbH, Austria at ~ 1 ppmv; with stated
accuracy better than 8 %). The calibration components were methanol, acetonitrile, acetaldehyde, ethanol,
acrolein, acetone, isoprene, crotonaldehyde, 2-butanone, benzene, toluene, o-xylene, chlorobenzene, α-pinene,
and 1, 2- dichlorobenzene. The background measurements were performed using a dry zero air cylinder every
two weeks. The raw data were processed using the Tofware post processing software (version 2.5.11,
TOFWERK AG, Thun, Switzerland) with the PTR module as distributed by Ionicon Analytik GmbH
(Innsbruck, Austria), running in the Igor Pro 6.37 environment (Wavemetrics Inc., Lake Oswego, OR, U.S.A.).
Volume mixing ratios (in ppbv) were calculated based on the method described by de Gouw and Warneke
(2007) and the literature reaction rates (*k*) of the ion with the $H_3O^+$ ion were applied when available (Cappellin
et al., 2012). For ions where the reaction rate had not been measured, a rate constant of $2 \times 10^{-9}$ cm$^3$ s$^{-1}$ was
assumed.
The mixing ratio of $NO_x$ was measured by chemiluminesence using the Serinus 40 Oxides of Nitrogen analyzer
(Ecotech) at IITD and a Model 42i (TEC, USA) at MRIU. The CO mixing ratio was analyzed with infrared
radiation absorption method using a CO Analyzer (Serinus 30, Ecotech) at IITD and a Model 48i (TEC, USA) at
MRIU.
**2.3 Source apportionment**
Source apportionment of VOCs was performed using the positive matrix factorization (PMF) receptor model
(Paatero and Tapper, 1994), which represents the measured PTR-ToF-MS mass spectral time series as a linear
combination of static factor profiles (characteristic of particular sources and/or atmospheric processes) and their
time-dependent concentrations. This can be represented in matrix notation as follows:
$$\mathbf{X} = \mathbf{GF} + \mathbf{E} \tag{2}$$
Where $\mathbf{X}$, $\mathbf{G}$, $\mathbf{F}$, and $\mathbf{E}$ are matrices corresponding to the measured mass spectral time series (time x *m/z*), factor
time series (time x factor), factor profiles (factor x *m/z*), and model residuals (time x *m/z*), respectively. In this
study, we used a total of 158 ions at IITD and 90 ions at MRIU measured by the PTR-ToF-MS (list of ions
utilized are shown in Table S1 and Table S2). Equation (2) is solved by minimizing the objective function, $Q$
(Eq. 3), using a weighted least-squares algorithm:
$$Q = \sum_i^n \sum_j^m \left(e_{ij}/s_{ij}\right)^2 \tag{3}$$
Here, $e_{ij}$ is the residuals (elements of **E**) and $s_{ij}$ represents the corresponding measurement uncertainty.
As recommended by Paatero (2003), ions with a signal-to-noise ratio (SNR) lower than 0.2 were excluded from
the input matrix, and ions with a signal-to-noise ratio (SNR) between 0.2 and 2 were down-weighted by
increasing their uncertainties by a factor of 2. In addition, in this study, a few ions such as methanol,
acetaldehyde, acetone, and acetic acid with mixing ratios 3-4 times higher than the other ions were excluded
from the input. Due to extremely high SNR compared to other compounds, inclusion of these ions in PMF leads
to solution where only these ions are well-explained, and useful source information is not retrieved.
PMF was implemented as the Multilinear Engine (ME-2) (Paatero, 1999), with the Source Finder (SoFi) toolkit
for Igor Pro (Wavemetrics, Inc., Portland; Canonaco et al., 2013) used for model configuration and post-
analysis. As different combinations of **G** and **F** can yield mathematically similar solutions (i.e., similar $Q$), ME-
2 enables intelligent rotational control to achieve reasonable solutions by involving constraints and external
data. In this study, constraints were applied combining a scalar $a$ (usually between 0 and 1) and a reference
profile (Canonaco et al., 2013). The $a$-value determines the extent to which the resolved factors $f'$ and $g'$ are
allowed to vary from the input reference elements of $f$ and $g$ according to Eq. 4:
$$f' = f \pm a \cdot f \quad \text{and} \quad g' = g \pm a \cdot g \tag{4}$$
**3 Results and discussion**
**3.1 Temporal and spatial variation**
Figure 2 presents time series of the mixing ratios of CO and $NO_x$ as well as of the sum of all VOCs analyzed in
the PMF model and temperature at the two sites. The fractional composition of VOCs in terms of the chemical
formula-derived families is included as well. The mean mixing ratios of CO were 1.29 ppmv, and 0.95 ppmv at
IITD and MRIU, respectively. The average mixing ratio of $NO_x$ at IITD was 148.0 ppbv, 5 times higher than at
MRIU (29.6 ppbv). Similarly, the summed VOC mixing ratio (i.e., all the ions included in PMF analysis, see
Table S1 and S2) was 27.6 ppbv at IITD, significantly higher than at MRIU (19.4 ppbv). The ambient
temperature showed large variations during the campaign with an overall increasing trend after 21 February.
Thus, the whole study period can be divided into two separate periods of relatively cold and warm temperatures.
Fig S1 shows the box-whisker plots of temperature during the two periods at the two sites. As shown in Fig. S1,
the average temperature was 17 ºC during the cold period and 23 ºC during the warm period at IITD. At MRIU,
the average temperature was very similar when VOCs were measured, with the cold days averaging at 16 ºC and
the warm periods at 23 ºC. During the warm period, the mean temperature reached the minimum at 6:00 LT, 1
hour earlier than in the cold period. These differences in temperature along with other meteorological
parameters are expected to have an impact on the emission profiles (e.g. Sekimoto et al., 2018) as well as the
dilution and chemical transformation processes.

To investigate temporal and site-dependent changes in the relative composition of the VOCs, the measured signals of different ions were classified into seven families based on their identified chemical formula, namely aromatic $C_xH_y$, non-aromatic $C_xH_y$, $C_xH_yO_1$, $C_xH_yO_2$, $C_xH_yO_3$, $C_xH_yN$, and $C_xH_yNO_z$ compounds. Aromatic $C_xH_y$ were classified due to their low H:C ratio, and oxidized aromatics are classified into the three $C_xH_yO_z$ families. Aromatic $C_xH_y$ were the largest fraction at both sites constituting about 45.4% at IITD (averaged 13.9 ppbv) and 34.3% at MRIU (averaged 7.0 ppbv). The high fractions of aromatic compounds indicate strong influences from anthropogenic emissions, mainly vehicle exhaust, at both sites. At IITD, the contributions of $C_xH_yO_1$ (21.9%, 5.6 ppbv on average) and $C_xH_yO_2$ (13.6%, 3.6 ppbv) were also significant, followed by non-aromatic $C_xH_y$ (11.3%, 3.5 ppbv), $C_xH_yN$ (5.2%, 1.2 ppbv), $C_xH_yO_3$ (2.3%, 0.6 ppbv), and $C_xH_yNO$ (0.4%, 0.5 ppbv). At MRIU, the VOC family composition was similar, with the exception of a higher fraction and concentration of non-aromatic $C_xH_y$ (23.2%, 4.2 ppbv), which was dominated by high $C_5H_8$ and $C_6H_{10}$ in the daytime compared to that at IITD. Besides, the averaged mixing ratios of the majority of the families were lower at MRIU than at IITD, except for non-aromatic $C_xH_y$ and $C_xH_yO_2$. The averaged mixing ratios of $C_xH_yO_1$, $C_xH_yO_2$, $C_xH_yN$, $C_xH_yO_3$, and $C_xH_yNO$ were 3.6 ppbv, 3.4 ppbv, 0.5 ppbv, 0.2 ppbv, and 0.09 ppbv, respectively.

Owing to the variation of planetary boundary layer height (PBLH), as well as emission sources, temperature, and solar radiation, the time series of the mixing ratios of VOCs, $NO_x$ and CO exhibited substantial differences between day and night periods. Fig. S2 shows the diurnal patterns of $NO_x$, CO, and different VOC families at both sites. The nocturnal mixing ratios of most of the VOC families, as well as those of CO and $NO_x$ were higher than during daytime, with much greater diurnal variation at IITD than at MRIU. The spatial difference in the diurnal variation may be due to a lower influence of local emissions at the suburban MRIU site because of lower population density and fuel consumption. In addition, the relative proportions of the VOC families varied over time, indicating different emission patterns and oxidation chemistry. For instance, substantial contributions and concentrations of aromatic compounds and $C_xH_yNO_1$ were observed at night at IITD, indicating strong anthropogenic emissions in the urban area, such as traffic-related emissions and solid fuel combustion. Meanwhile, higher daytime contributions were found for the $C_xH_yO_1$ and $C_xH_yO_2$ families; and in particular the $C_xH_yO_3$ family peaked around midday at IITD, indicating tropospheric aging and secondary formation during daytime. Moreover, the nocturnal concentrations of all the gas phase species were higher at IITD compared to MRIU. During daytime, however, the concentrations of non-aromatic $C_xH_y$, $C_xH_yO_2$, as well as CO at MRIU were higher compared to those at IITD.

**3.2 VOC source apportionment**

**3.2.1 Solution selection**

We evaluated unconstrained solutions from 4 to11 factors individually for both sites, and solutions were selected based on interpretation of the spectra, reference information from emission inventory, correlation of VOC factors with ancillary measurements, and mathematical diagnostics describing PMF performance. A six-factor solution was selected as the best representation of the data measured at both sites, which includes two traffic factors, two solid fuel combustion factors, and two secondary factors. An industrial factor is not presented in the results, as marker compounds typically originating from industrial emissions are either not measured

(such as halogen compounds) nor included in this PMF analysis (such as methanol and acetone as described in
Section 2.3). In terms of PMF performance, the $Q/Q_{exp}$ did not show any step change with increasing factor
number (Fig. S3). Solutions with less than six factors failed to provide reasonably separated sources, while
solutions with more factors led to additional traffic or solid fuel combustion factors that did not improve the
interpretability of the results. These additional factors were likely the result of minor variations across discrete
instances of a given source (e.g. vehicle-to-vehicle variations in VOC composition) and slight differences in
reactive history.
At IITD, the concentrations of several aromatics were very high, being 10-20 times higher than those of the
major phenols, and over 100 times higher than for the compounds with the lowest concentration. As a result,
initial PMF results were over-weighted towards explaining aromatic variability. Thus, aromatics were
apportioned to all the resolved factors, including factors that otherwise appear secondary and where such high
attribution is not reasonable. To this mathematical artifact and guide the model towards environmentally
reasonable results, constraints were applied to the IITD secondary factors as follows. First, PMF was performed
with a new input matrix excluding the most abundant aromatic ions, namely $C_6H_6H^+$, $C_7H_8H^+$, $C_8H_{10}H^+$, and
$C_9H_{12}H^+$ (PMF results with the new input are shown in Fig. S4). This yielded a reasonable five-factor solution,
from which the spectra of the two secondary factors were selected for use as reference profiles in the original
dataset. PMF was then executed again on the full dataset (i.e., all 158 ions including the aromatics), with
constraints applied to the two secondary factors using the $a$-value approach. Since we obtained the reference
from the same dataset, the $a$-value was set to a smaller range than the usual case. Here, solutions with the $a$-
value ranging from 0.1 to 0.3 were evaluated and no significant differences in terms of the spectra and temporal
variations were observed among all the results. In the following analysis, the solution with an $a$-value=0.1 for
both factors was selected as the IITD result, and the result with an $a$-value=0.3 is presented in Figure S5.
At MRIU, the lower aromatic concentrations meant that that even the unconstrained PMF did not apportion
significant aromatic mass to resolved secondary factors. For consistency, we tested also the constraint-based
method described above, but the two methods did not yield significant differences at MRIU (see Figure S6).
Therefore, at MRIU the unconstrained solution was selected for further interpretation.

**3.2.2 Factor identification**

Figures 3 and 4 show the factor profiles (a), time series (b), and diurnal patterns (c) of the selected PMF solution
at IITD and MRIU, respectively. In the following, we present a detailed discussion of the factor characteristics
at IITD. The MRIU factors are qualitatively similar and therefore a detailed discussion is not repeated, but rather
a site comparison is presented in Section 3.3.
The first two factors are related to vehicle emissions and denoted Traffic1 and Traffic2. Both are enhanced
during the night and relatively low during daytime. The mass spectra of both Traffic1 and Traffic2 are
dominated by aromatic $C_xH_y$ compounds, namely $C_6H_6$ (*m/z* 79.054), $C_7H_8$ (*m/z* 93.070), $C_8H_{10}$ (*m/z* 107.086),
$C_9H_{12}$ (*m/z* 121.101), and $C_{10}H_{14}$ (*m/z* 135.117). These ions are tentatively attributed to benzene, toluene, C8-
aromatics, C9-aromatics, and C10-aromatics, which are well-studied markers for vehicular emissions (Yao et
al., 2015; Cao et al., 2016). While Traffic1 is mostly composed of pure aromatics, Traffic2 includes relatively
high contributions from non-aromatic hydrocarbons and some oxygenated compounds. As shown in Fig. 3 (b),
both traffic factors exhibit temporal variations similar to $NO_x$, which originates mainly from vehicle emissions
in an urban area. The correlation of Traffic 2 with $NO_x$ ($R^2$=0.76) is better than either Traffic1 with $NO_x$
($R^2$=0.55) or their sum (Traffic1+Traffic2) with $NO_x$ ($R^2$=0.67). To alleviate urban traffic congestion, heavy-
duty vehicles are banned in Delhi during the rush hours and daytime (7:00-21:00 LT). As a result, most heavy-
duty traffic in Delhi occur overnight rather than during daytime. This difference in traffic patterns corresponds
to the temporal differences in Traffic1 and Traffic2. As shown in Fig. S7, the ratio of Traffic 2 to Traffic 1 is
very low during the day time and starts to increase slightly from 16:00 LT. Although both Traffic1 and Traffic2
are high during the night with their maximum concentrations around 21:00 LT, the Traffic2/Traffic1 ratio is as
low as 0.6 at that time. The ratio increases overnight with a sharp increase during the early morning, reaching a
maximum value of 1.4 at 07:00 LT, suggesting that Traffic1 is the dominant traffic source both day and night.
This can be explained as cold start emissions from gasoline vehicles emit high amounts of VOCs while heavy-
duty vehicles contribute much less to VOCs but more to BC and $NO_x$ (Platt et al., 2017). Further, the spectrum
of Traffic2 is characterized by high fractions of high mass aromatic compounds. For instance, the ratio of C8/C7
aromatics is tripled in Traffic2 compared to that in Traffic1, which is similar to the ratio of the emission factors
from previous studies (3.5 times (Gentner et al., 2013)). Since the vehicle number in Delhi has increased
dramatically in the past two decades (the registered vehicular population has tripled since 1994 and has reached
7.6 million), the related pollution is considered as an increasingly significant source of atmospheric pollution.
Overall, the sum of the two traffic factors is found to be the dominant VOC source in Delhi, with contributions
of 33.8% from Traffic1 and 22.8% from Traffic2 to the total analyzed VOCs.
Two other factors are found to be related to solid fuel combustion (SFC) and are named SFC1 and SFC2. SFC1
is characterized by high loadings of aromatics, as well as oxygenated ions, such as $C_6H_6(C_nH_{2n})O_1$,
$C_6H_6(C_nH_{2n})O_2$, $C_4H_4(C_nH_{2n})O_1$, $C_4H_4(C_nH_{2n})O_2$, and $C_5H_4O_{1-2}$. These ions are tentatively attributed to phenolic
compounds and furans (Stockwell et al., 2015; Bruns et al., 2017). The relative fractions of emission factors
(EF, in mg per kg fuel) of $C_5H_4O_2$ (100 mg kg$^{-1}$), $C_6H_6O$ (110 mg kg$^{-1}$), and $C_6H_6O_2$ (60 100 mg kg$^{-1}$) are
similar to their relative concentrations in SFC1 (i.e. $C_5H_4O_2$ (m/z 97.028, 0.019), $C_6H_6O$ (m/z 95.049, 0.021),
and $C_6H_6O_2$ (m/z 111.044, 0.012)), consistent with the identification of the factor. Besides, previous studies
have reported that furans and phenols are emitted in high quantities by wood/biomass burning and coal
combustion (Bruns et al., 2017; Klein et al., 2018) and thus are regarded as important markers for these
combustion processes. In urban Delhi, the use of solid fuels is not limited to wood and coal, but includes open
combustion of many types of biomass and even waste. In addition, traditional stoves remain popular in some
residential households and restaurants, and their VOC emission profiles are still not clear. SFC1 is identified as
being more related to primary emissions from solid fuel combustion, which might include many types of
biomass burning, coal combustion, and even trash combustion. The spectrum of the SFC2 factor is dominated
by benzene and to a lesser extent toluene, while the more reactive phenols and furans are only present in small
amounts, suggesting increased age. In addition, it includes a high mass fraction of N-containing compounds,
such as $C_3H_3N$, $C_4H_5N$, and $C_7H_5N$. Although the chemical identification of these ions is not confirmed, these
ions were reported in biomass burning as well (Stockwell et al., 2015; Sekimoto et al., 2018) and may be
attributed to nitriles, which have longer OH lifetimes compared to furans and phenols (Meylan and Howard,
1993; Atkinson et al., 1989). SFC2 also explains a higher fraction of higher molecular-weight molecules, e.g.
$C_6H_5NO_3$ and $C_7H_7NO_3$. These ions are attributed to nitrophenols, which are semivolatile compounds found in
biomass burning-related secondary organic aerosol (Mohr et al., 2013). As shown in Fig 3(c), SFC1 exhibits a
diurnal pattern with a maximum value at 20:00 LT and a smaller peak at 08:00 LT, consistent with residential
heating and/or cooking activities. SFC2 is also high at night, although it peaks several hours later than SFC1 (at
5:00 LT), and decreases strongly during daytime without any morning peak. This suggests that the nocturnal
peak of SFC2 may be due to dark oxidation of emitted precursor VOCs, e.g. by nitrate radicals. Therefore,
considering emissions, chemical transformation and reactivity, SFC1 represents more primary emissions from
various types of solid fuel combustion in central Delhi, while SFC2 is associated with more aged emissions
from solid fuel combustion.
Finally, two secondary VOC factors are identified, the spectra of which are distinguished from the other factors
by a strong presence of oxygenated compounds. Secondary VOC1 (denoted SecVOC1 in the following)
contributes a large fraction of $C_2H_4O_3$ and $C_4H_2O_3$, and the spectrum is dominated by $C_4H_8O$, $C_3H_4O_2$, $C_3H_6O_2$,
and $C_3H_6O_3$, although the relative contributions to these ions are comparably lower than to $C_2H_4O_3$ and $C_4H_2O_3$.
These ions are likely oxidation products from various photochemical processes and are not unique to specific
precursors and oxidation processes. The time series of SecVOC1 follows that of the solar radiation, which has a
regular contribution cycle during daytime. The diurnal of SecVOC1 shows a rapid enhancement starting from
around 7:00-8:00 LT and declines continuously after 13:00 LT. This indicates that while many of these ions can
be formed rapidly during daytime, they may have a short lifetime owing to partitioning to the condensed phase
and/or heterogeneous processes. Secondary VOC2 (SecVOC2) has significant contributions from $C_3H_4O_2$ and
$C_3H_6O_2$ as well, but the relative contributions to $C_3H_4O_2$, $C_4H_6O_{2-3}$, $C_6H_6O_2$, $C_8H_8O_3$, and $C_6H_5NO_3$ are
substantially higher than for SecVOC1. It also includes many oxygenated compounds with higher molecular
weight, e.g. $C_5H_{10}O_{1-2}$, $C_6H_{10}O_{1-2}$, and $C_6H_{12}O_{1-2}$. SecVOC2 (Fig. 3(c)) is not only high during daytime, with a
slow rise in the morning and a peak around 10:00-12:00 LT, but is also high at night, reaching its nocturnal
maximum from 20:00 LT to midnight. Major fractions of alkyl nitrates ($RONO_2$) are detected as ($ROH \cdot H^+$)
fragment ions by the PTR-ToF-MS (Aoki et al., 2007). Therefore, species such as $C_5H_{10}O_{1-2}$, $C_6H_{10}O_{1-2}$, and
$C_6H_{12}O_{1-2}$, may be partially attributed to fragments from organic nitrates, which contribute to the high nocturnal
SecVOC2 mixing ratio. Therefore, it is likely that SecVOC1 is a mix of first generation products and later-
generation oxidation products, while SecVOC2 is possibly associated with secondary generation products and
nighttime chemistry.
**3.3 Comparison of VOC sources**
Figure 5 compares the diurnal cycles and mass spectra of the two Traffic and the two SFC factors between the
two sites. While the factor profiles are similar for the primary factors (i.e. Traffic1, Traffic 2 and SFC1 at both
sites share the same major ions) their diurnal patterns are different. Higher nocturnal concentrations and day-
night variations are observed at IITD than at MRIU for all three primary factors. For example, for Traffic1 the
nocturnal maximum is 5.3 times higher than the daytime minimum at IITD and only 2.7 times higher at MRIU.
For SFC1, the corresponding numbers are 36.8 at IITD and 6.9 at MRIU. This is possibly due to increased
traffic density and more intense combustion activities in the populated urban areas. Besides, Fig. S8 presents
concentration weighted trajectory (CWT) plots of the six factors at both sites. Details of the back-trajectory and
CWT analysis are shown in the Supplement. As shown in Fig. S8, high concentrations of primary factors are
found both north and northwest of IITD, whereas high primary factors originate from southeast of MRIU,
different from the direction of IITD.
Unlike the primary factors, the mass spectra of SFC2 differ between the two sites. At IITD, SFC2 is not only
rich in benzene, but also in toluene. However, as shown in Fig. 5, toluene is much lower at MRIU, while the
relative benzene concentration remains high. In addition, SFC2 at MRIU is richer in some oxygenated
compounds, such as $C_6H_{12}O_{1-2}$, $C_5H_{10}O_2$, $C_6H_{10}O_2$, $C_4H_{4/6/8}O_2$. These ions are the major ions found in the
SecVOC2 factor at IITD. Because toluene has a much shorter photochemical lifetime than benzene, this
suggests that SFC2 comprises less freshly emitted but more aged compounds at MRIU than at IITD. Although
the SFC2 factors exhibit similar diurnal trends at the two sites, the time of daily maximum is different. At
MRIU, SFC2 reaches the maximum at 07:00 LT, which is approximately 2 hours later than that at IITD.
Further, the maximum averaged SFC2 mixing ratio at MRIU is 4.41 ppbv, lower than the peak value of 6.20
ppbv at IITD, and consistent with the higher primary emission levels at IITD.
The two secondary VOC factors, in particular, show major differences between the two locations in terms of
diurnal variations and spectra. On a simple level, the two SecVOC2 are considered to represent related factors
because of their similar long-term trends with local temperature (Fig. 3 and 4). However, the mass spectra are
different, indicating different aging mechanisms. As illustrated in Fig. 3 and 4, the spectrum of SecVOC1 is
dominated by $C_4H_2O_3$ and $C_2H_4O_3$ at IITD but less so at MRIU for both SecVOC factors, especially in the case
of $C_4H_2O_3$. Further, the IITD SecVOC1 shows additional origins apart from the directions of primary factors,
indicating influences from transport (Fig. S8). The SecVOC2 at IITD, however, is comparable to the MRIU
SecVOC1 in terms of factor fingerprint (Fig. S9). The spectra of the two factors are both characterized by high
contributions to $C_3H_4O_2$, $C_4H_6O_{2-3}$, $C_5H_{10}O_{1-2}$, $C_6H_{10}O_{1-2}$, and $C_6H_{12}O_{1-2}$, with a relatively higher contribution to
$C_4H_2O_3$ and $C_2H_4O_3$ by IITD SecVOC2. However, their time trends and diurnal variations are rather different.
Although they both increase during daytime, the maximum is found about 4 hours later at MRIU than at IITD,
possibly due to less primary emission in the morning. The elevated periods of IITD SecVOC2 are not only
related to higher temperature, but also consistent with the periods with much stronger primary emissions (e.g.
Traffic factors). Similarly, the time series of MRIU SecVOC1 is also high in January when all the primary
factors are high compared to the rest of the periods. Besides, as shown in Fig. S8, SecVOC2 at IITD and
SecVOC1 at MRIU exhibit similar geographical origins as the primary factors at the respective sites, suggesting
local oxidation of primary emissions. The spectrum of MRIU SecVOC2 is also loaded with $C_4H_2O_3$, $C_2H_4O_3$, as
well as high $C_4H_8O$ and $C_3H_4O_2$, but the contributions to these ions are much lower than SecVOC1. In addition,
MRIU SecVOC2 exhibits a similar time series as that of local temperature, which increases in the warmer
period of the campaign. The overall trend is quite different from the local primary factors, but has some
similarities with that of IITD SecVOC2. As shown in Fig. S8, high concentrations of MRIU SecVOC2 originate
mainly from north and northwest directions, consistent with the location of the IITD site. Therefore, it is
possible that MRIU SecVOC2 represents oxidized VOCs on a relatively regional scale.
Figure 6 presents the stacked factor diurnal patterns at the two sites. Much higher nocturnal mixing ratios of
VOCs are observed at IITD than at MRIU, leading to pronounced day-night differences, while during daytime
higher mixing ratios are found at MRIU than at IITD (with 15 ppbv at MRIU in the mid-afternoon, compared to
10 ppbv at IITD). This can be explained by the fact that at IITD the VOC mixing ratio is dominated by primary
emissions, especially during the night, while at MRIU the VOC mixing ratio is dominated by secondary
formation, at least during the day. At IITD, primary factors contribute to around 85% before midnight, with
large fractions coming from Traffic factors and SFC1. Afterwards, SFC2 gradually increases and contributes to
over 40% of the total mixing ratio before sunrise, when other primary factors decrease to about 50%. At the
suburban MRIU site, although the nocturnal mixing ratio of primary VOCs are about half as high as that at
IITD, they contribute to around 70% of the total VOCs concentrations. In the daytime, however, the SecVOC
contribution is as high as 54%, compared to the maximum of 41% at IITD. Generally, the MRIU SecVOC
contributes about 15-20% more than that at the IITD site throughout the day, mostly due to increased SecVOC1.
This may be partially explained by the differences in rotation techniques we applied in running the model for the
two secondary factors. As illustrated in Figure S10, by applying the similar $a$-value approach to MRIU as used
at IITD, the constrained MRIU SecVOC is around 1.5 ppb lower than the raw PMF result, and contributes a
maximum of 10% less at around 14:00-15:00 LT. However, SecVOC contributes 31% to the total analyzed
VOCs mixing ratio in the constrained MRIU solution, instead of a contribution of 34% from the unconstrained
result as illustrated above. Both PMF solutions resolved at MRIU show much higher fractions of SecVOC
compared to that at IITD (16%). Therefore, the rotation techniques only play a minor part on the discrepancies
between the two sites. More importantly, the difference in SecVOC is probably due to different oxidation
conditions at the two sites. Owing to the high mixing ratio of $NO_x$ and the suppression of oxidants like OH
radicals, the chemical oxidation of primary VOCs occurs to a greater extent downwind of urban emission
sources. As shown in Fig. S8, high mixing ratios of SecVOC at MRIU originate from northwest directions,
different from the origins of local primary factors (southeast). Besides, the longer oxidation time may be another
reason of the higher SecVOC mixing ratios at MRIU. In addition, several ions below $m/z$ 60 that can be detected
by the PTR-ToF-MS are excluded from PMF as discussed in Sec. 2.3, such as methanol, acetaldehyde, acetone
and acetic acids. These ions are 3-4 times higher than the dominant ions in the PMF analysis, possibly owing to
much higher emission rates and natural abundance. Besides, other excluded compounds such as $C_1$-$C_4$ alkanes
and $C_1$-$C_4$ alkenes which are not detectable by the PTR-ToF-MS are substantial contributors to the total VOC
mixing ratio as well. However, these ions are minor contributors to SOA formation and only substantially
contribute to the formation of ozone, which is a major issue in summer. Although the mixing ratio of the sum of
VOCs in the PMF only accounts for 39.6 % at IITD and 24.2 % at MRIU (Fig. S11), many of these compounds
are the dominant precursors in terms of SOA formation (Wu and Xie, 2017, 2018).

**3.4 Evaluation of biogenic signatures**

Biogenic VOCs, i.e. isoprene and monoterpenes, are not separated into a specific factor in this study. This is in
part due to the small number of ions that can be unambiguously assigned to these sources. Specifically, the
structure assignment and quantification of non-aromatic $C_xH_y$ can be uncertain not only because of isomers, but
also because of fragments from aldehydes, alcohols, and other long-chain aliphatic hydrocarbons. Therefore,
$C_5H_8$ ($m/z$ 69.070) and the sum of $C_{10}H_{16}$ ($m/z$ 137.132) and $C_6H_8$ ($m/z$ 81.070, a major fragment of $C_{10}H_{16}$) can
only serve as upper estimates of the actual isoprene and monoterpenes, respectively. The averaged mixing ratios
of isoprene and monoterpenes are low during the campaign, with 0.8 ppbv and 0.46 ppbv, respectively at IITD,
and 1.2 ppbv and 0.32 ppbv, respectively at MRIU. Figure 7 illustrates the explained variation of the two major
biogenic markers (i.e. isoprene and monoterpenes) and several other ions possibly related to their oxidation
products. Among them, $C_4H_6O_{1-2}$ (*m/z* 71.049, *m/z* 87.044) and $C_5H_8O_{1-2}$ (*m/z* 85.065, *m/z* 101.060) are found in
the photooxidation of isoprene, and $C_9H_{14}O$ (*m/z* 139.112) and $C_{10}H_{14/16}O$ (*m/z* 151.112, *m/z* 153.127) are
potential photochemical products from monoterpenes. Instead of being related to a specific secondary factor, the
biogenic markers are largely explained by primary factors. There are several possible explanations. First, both
isoprene and monoterpenes have been found in biomass burning emissions from numerous laboratory and field
studies (e.g. Bruns et al., 2017). Second, previous studies also indicated anthropogenic sources of isoprene from
vehicular emissions (Borbon et al., 2001;Wagner and Kuttler, 2014). One recent paper showed that fragrances
and personal care products may be an important emission source of urban monoterpenes and are correlated with
traffic emissions (McDonald et al., 2018). Third, considerable amounts of these ions may be affected by isomers
and fragments of larger ions originating from primary emissions. Finally, the two BVOCs are very reactive
compounds, which have very short lifetimes especially during daytime, and also during nighttime if $NO_3$ is
present. Therefore, large amounts of BVOCs can be degraded very quickly via photochemical oxidation during
daytime, resulting in lower daytime BVOC mixing ratios despite of stronger natural emissions. As shown in
Figure S12, both isoprene and monoterpene are high at night and very low during daytime, similar to the diurnal
trends of the primary factors. The only exception is isoprene at MRIU, which may be due to strong daytime
emissions or contributions from isomers/fragments.
As shown in Figure 7, the $C_xH_yO_z$ ions are explained partially by secondary factors, indicating possible
secondary formation from biogenic precursors. However, at both sites, considerable amounts of these ions are
also explained by emissions from SFC. Previous studies showed that the emission factors of $C_4H_6O_{1-2}$ are
comparable to that of benzene (200 mg kg$^{-1}$), and the EFs for $C_5H_8O_{1-2}$ and $C_{10}H_{16}O$ are comparable to that of
toluene (27 mg kg$^{-1}$) in biomass/wood burning emissions (Bruns et al., 2017; Koss et al., 2018). Besides, as
these small molecular weight ions are possible products from various precursors and chemical pathways, their
structures and precursor definitions are uncertain. However, the explained variations can still provide
information on the aging processes. In general, the fraction explained by secondary factors increases as the ions
become more oxygenated. In addition, at IITD SecVOC1 explains much higher fractions of the $C_xH_yO_1$ ions
compared to SecVOC2, while the $C_xH_yO_{2-3}$ ions are to a great extent explained by SecVOC2. As some of these
less oxygenated ions originate from both fast photochemical formation and later-generation production, they are
explained by both secondary factors. For example $C_4H_6O$, which can be attributed to MVK (methyl vinyl
ketone) and MACR (methacrolein), is formed rapidly via isoprene OH oxidation, but can also be found as
second generation products via several other oxidation pathways. Generally, both biogenic markers and their
oxidation products are low during the measurement period, and these ions contribute only small amounts to the
two secondary factors, indicating minor contributions from biogenic emissions at the two sites during the
campaign.
**3.5 Characteristics of selected ions**
Figure 8 shows the diurnal patterns of the explained variation of each PMF factor to selected ions measured at
the two sites, stacked such that the height is the total modelled concentration. The red line in each plot
represents the mean diurnal pattern of the measured mixing ratio. The model represents over 80% of most of
these selected ions, and thus in general explains the variations of these ions well. The comparison of these
diurnal patterns illustrates that distinct differences exist in the mixing ratios, variations, and source compositions
of certain VOCs. The assignment of a specific compound structure to each ion can be ambiguous, particularly
for the spectra obtained from ambient air due to the possibility of isomers and/or fragmentation. Still, we
provide tentative candidates for the nine ions based on their molecular formulas and compounds found in
previous fuel combustion and oxidation simulation studies. Structural assignments for aromatics and PAHs are
more certain because the low H to C atomic ratio reduces the number of reasonable structures, therefore, $C_6H_6$,
$C_7H_8$ and $C_{10}H_8$ are mainly attributed to benzene, toluene, and naphthalene, respectively. Also included in Fig. 8
are $C_5H_4O_2$, $C_6H_6O$, $C_2H_4O_3$, $C_4H_2O_3$, and $C_6H_5NO_3$; we suggest that the most likely structural assignments are
furfural, phenol, a fragment of peroxyacetyl nitrate (PAN), maleic anhydride, and nitrophenol, respectively.
Benzene and toluene are well-interpreted aromatics, which contribute significantly to emissions from several
combustion processes, including vehicle emissions, biomass burning, coal combustion, etc. (Pieber et al., 2018;
Bruns et al., 2017; Klein et al., 2018). As shown in Fig. 8 (a), at IITD benzene originates to a significant extent
from traffic emissions (53% on average) over the whole day and in addition from the SFC factors (47% on
average), largely at night. The traffic fraction is lowest (29%) in the early morning and increases during
daytime, with a maximum of 74 % around 17:00-18:00 LT, with the opposite trend for the SFC fraction. This
suggests that the IITD site is dominated by pollution from vehicular emissions because of the highly trafficked
streets nearby, and also influenced by strong solid fuel combustion due to the high population density. At
MRIU, however, there are comparable benzene contributions by the Traffic1, SFC1, and SFC2 factors. Even
though the sum of the SFC mixing ratios is lower at MRIU compared to IITD, the relative SFC contribution is
higher at MRIU. Although the traffic fraction increases during daytime at both sites, it explains a maximum of
74% at IITD and 49% at MRIU around 17:00-18:00 LT, with the opposite behavior for the SFC fraction.
Toluene displays a similar diurnal cycle and source contributions as benzene, characterized by significant
contributions from primary sources, especially Traffic1. The traffic fraction is as high as 90 % around 18:00 LT
and reaches a minimum of 67 % around 10:00 LT at IITD, which is much higher than that of benzene. The
benzene/toluene (B/T) of the traffic factor ranges from 0.34 to 0.40, which is comparable to the emission factors
of benzene/toluene for gasoline emissions (0.58, Gentner et al., 2013). Besides, the traffic fraction is lowest in
the morning, which is associated with the strong emissions of SFC in the morning and high traffic emissions
during the rush hour in the late afternoon. The B/T is 3.4 for the SFC factors, which is in the reported range of
2-7 for residential biomass/wood burning (Bruns et al., 2017; Koss et al., 2018). Similarly, traffic is also the
highest source of toluene at MRIU, ranging from 67%-77%.
As mentioned above, phenolic compounds and substituted furans are found to be significant oxygen-containing
compounds at the two sites. Furfural is an important marker from biomass burning, which arises from pyrolysis
of cellulose and hemicellulose. Phenols are detected in biomass burning and coal combustion emissions as the
most abundant oxygenated aromatic compounds. (Stockwell et al., 2015; Klein et al., 2018; Bruns et al., 2017)
Indeed, as shown in Fig. 8 (c,f), the majority of furfural and phenol are explained by the two SFC factors at both
sites. The two SFC factors contribute 91% to furfural and 85% to phenol at IITD, while the corresponding
contributions are 70% and 57% at MRIU. SFC1 contributes around 54% and 39% during the night and morning,
respectively, with a higher night-to-day ratio at IITD, which captures both the characteristics of residential
heating styles and urban/suburban usage differences due to population density. The SFC2-to-SFC1 ratio
generally increases during the night. As described earlier, one possible explanation is that SFC2 results from the
fast nocturnal evolution of primary SFC1 emissions probably driven by night chemistry under high $NO_x$
conditions (Tiitta et al., 2016). In addition to the SFC factors, at IITD around 8% of furfural and 4% of phenol
are explained by the two SecVOC factors, compared to around 11% and 25% at MRIU, respectively.
Nitrated phenols are important biomass burning tracers, in particular for secondary aerosol formation (Mohr et
al., 2013; Bertrand et al., 2018). Here, the diurnal cycles of $C_6H_5NO_3$ at the two sites illustrate contributions
from both emission and atmospheric transformation. At both sites, SFC2 accounts for a large percentage of the
nocturnal nitrophenol mixing ratios while the SecVOC factors explain the remaining fraction. This is consistent
with SFC2 representing nighttime aging of SFC emissions. It is worth noting that during the night there is a
much larger unexplained fraction (around 35%) at IITD than at MRIU, with a corresponding uncertainty in the
source contributions. Still, the main difference at the two sites relates to the contributions of the SecVOC
factors. At IITD, the SecVOC2 factor contributes more at night than during daytime, probably due to nocturnal
formation including $NO_3$ chemistry and the photolysis reactions of $C_6H_5NO_3$ during daytime. At MRIU, both
SecVOC1 and SecVOC2 contribute to $C_6H_5NO_3$, indicating the combined influence from local and regional
chemistry.
Maleic anhydride and PAN display distinct variation patterns at the two sites. Maleic anhydride is the product
from the photo-oxidation of furans, unsaturated carbonyls, and aromatics (Bierbach et al., 1994; Yuan et al.,
2017). The mixing ratio of maleic anhydride is about 5 times higher at IITD than that at MRIU during daytime
and is only explained by SecVOC1. As shown in Fig S13, high mixing ratios of maleic anhydride originate from
north (primary emissions) and northwest of IITD, the same directions as SecVOC1. Similarly, high mixing
ratios of maleic anhydride at MRIU originate both from southeast (primary emissions) and northwest (IITD site)
directions. During nighttime, the maleic anhydride mixing ratio at MRIU is similar to the one at IITD, with
considerable amounts explained by SecVOC2 and SFC2. A recent study indicates that furans contributed to over
90% of maleic anhydride within the first 4 hours of OH oxidation in biomass burning plumes (Coggon et al.,
2019). Therefore, the higher morning peak and nocturnal mixing ratios of furans may be the driver of much
higher maleic anhydride at IITD. Besides, it is possible that under the conditions of high humidity and aerosol
mass loading, maleic anhydride could react and/or condense away before the city plume reaching the suburban
area. In contrast to maleic anhydride, $C_2H_4O_3$ shows a higher mixing at MRIU than at IITD. $C_2H_4O_3$ is mainly
attributed to a fragment of PAN, however, a small fraction is also related to primary factors at both sites (Fig. 8).
This ion was reported in laboratory studies on primary biomass burning emissions (Bruns et al., 2017). Possibly,
small amounts of this ion might be attributed to unknown isomers, which are directly emitted or rapidly formed
in combustion processes. PAN is formed in VOC oxidation with $NO_x$ (de Gouw and Warneke, 2007). Due to its
long lifetime at low temperature, PAN can undergo long-range transport and then play an important role in
tropospheric ozone formation in the remote areas when it decomposes (Fischer et al., 2014). As illustrated in
Fig. S13, high mixing ratios of $C_2H_4O_3$ originate from north and far northwest, indicating influence of both local
emission/oxidation formation and long-range transportation. At MRIU, however, high mixing ratios of $C_2H_4O_3$
mainly originate from northwest, suggesting the influence of transportation from the urban areas. Besides, PAN
is mainly explained by SecVOC1 at IITD, and a much higher PAN mixing ratio is observed at the suburban
MRIU, originating from both secondary VOC factors, indicating a longer oxidation time at this site.
**4 Conclusions**

Measurements of volatile organic compounds (VOCs) were performed simultaneously at two different sites in Delhi, India using two PTR-ToF-MS instruments. Much higher mixing ratios of VOCs were observed at the urban IITD site, with higher nocturnal contributions from anthropogenic emissions than at the suburban MRIU site. Using positive matrix factorization, we found a 6-factor solution at both sites, each consisting of two traffic factors, two solid fuel combustion factors, and two secondary factors. Anthropogenic activities were shown to be important VOCs sources at both sites. Among them, traffic related emissions comprised the dominant source, contributing 56.6% at the urban IITD site and 36.0% at the suburban MRIU site to the total analyzed VOC mixing ratio (excluding high-intensity VOCs as described in Section 2.3). Solid fuel combustion contributed 27.5% at the urban IITD site, and even 30.4% at the suburban MRIU site. Secondary formation was the most important source of VOCs during daytime, and contributed 15.9% at IITD and 33.6% at MRIU to the total concentration. Higher mixing ratios of oxygenated VOCs were found at the suburban site, likely due to suppression of oxidant levels in the urban atmosphere and longer aging time at the suburban site.

Comparison of the factor diurnals and profiles indicated that the anthropogenic sources were similar at two sites, even though the VOC mixing ratios at MRIU were much lower than IITD. The secondary factors, however, were different at the two sites both in terms of diurnal variations and of spectra. The two secondary VOC factors exhibited higher mixing ratios and contributions at the suburban site, in particular during daytime. At IITD, the two SecVOC factors indicated different aging process under strong primary emissions, while at the suburban MRIU they suggested local oxidation and regional aging. Besides, evaluation of biogenic markers indicated very small influence from biogenic emissions and their oxidation products.

This work highlights the crucial role that anthropogenic sources play in the pollution levels and variation characteristics of VOCs in the ambient atmosphere of Delhi, India. Further control measures to reduce emissions from traffic exhaust and solid fuel combustion are urgently needed to mitigate the severe pollution and the environmental impact of VOCs as well as aerosols (not investigated in this paper) in this region. Significant differences in the concentrations as well as the pollution sources stress the complexity of both emission and chemistry in this region. Long-term measurements of a set of VOCs, aerosols, and other oxidants would be ideal to obtain a more detailed understanding of the formation mechanisms during various conditions.

**Data availability**

Data will be available at a zenodo repository (at the time of publication).

**Acknowledgement**

This work was supported by the SNF projects 200021_169787 (SAOPSOAG), BSSGI0_155846 (IPR-SHOP), IZLCZ2_169986 (HAZECHINA), and 200021_162448/1 (China-XRF). SNT gratefully acknowledges the financial support provided by the Department of Biotechnology (DBT), Government of India to conduct this research under grant no. BT/IN/UK/APHH/41/KB/2016-17 dated 19[th] July 2017 and financial support provided by the Central Pollution Control Board (CPCB), Government of India to conduct this research under grant no. AQM/Source apportionment_EPC Project/2017.

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

**Figure captions:**

**Fig. 1** Maps of the study region and the two sampling sites (from Google Maps). The black circle denotes the urban site at IITD and the black square the suburban site at MRIU.

**Fig. 2** Temporal variations of CO, $NO_x$, analyzed VOC mixing ratios and temperature as well as the contributions of seven VOC families at the two sites.

**Fig. 3** PMF results at IITD, showing (a) factor profiles (b) time series and (c) diurnal patterns. In (a), the left axis for each factor profile is the relative composition of each factor (i.e., horizontal sum is 1) and the right axis is the relative contribution of each factor to a given ion. Ions are colored based on the seven family classes, as described in Section 3.1. (b) Temporal evolution of resolved factors at IITD, together with external reference data, e.g. NOx, CO, solar radiation and temperature. (c) Medians of diurnal cycles of factors at IITD, shaded with interquartile ranges as well as the $10^{th}$ and $90^{th}$ percentiles.

**Fig. 4** PMF results at MRIU, showing (a) factor profiles (b) time series, and (c) diurnal patterns. (a) Relative composition (left axis) and relative contribution (right axis) of each factor to a given ion. Ions are colored based on the seven family classes, as described in Section 3.1. (b) Temporal evolution of resolved factors at MRIU, together with external reference data, e.g. $NO_x$, CO, solar radiation and temperature. (c) Medians of diurnal cycles of factors at MRIU, shaded with interquartile ranges as well as the $10^{th}$ and $90^{th}$ percentiles.

**Fig. 5** Comparisons of averaged factor diurnal patterns and factor profiles at the two sites. The left panels present the factor profiles with the IITD spectrum on top and the MRIU spectrum on bottom, color coded by the VOC families described in Section 3.1. The right panels show the box and whisker plots of diurnal cycles at the two sites, with the blue panel representing IITD and the red panel MRIU.

**Fig. 6** (a, c) Diurnal patterns of factor mixing ratios at the two sites, color-coded by the six retrieved factors. (b, d) Diurnal patterns of the fractional contributions of the factors at the two sites.

**Fig. 7** Explained variations of selected ions at the two sites, stacked such the sum is the total explained variation, color coded by the six factors (a) at IITD and (b) at MRIU. Missing ions at MRIU were excluded from PMF analysis due to low SNR. The possible candidates for these ions are isoprene ($C_5H_8$), MVK+MACR ($C_4H_6O$), 2,3-butanedione ($C_4H_6O_2$), 3-methyl-3-butene-2-one ($C_5H_8O$), methyl methacrylate ($C_5H_8O_2$), monoterpenes ($C_{10}H_{16}$), and camphor ($C_{10}H_{16}O$), respectively.

**Fig. 8** Diurnal cycles of selected marker ions, stacked with explained variations by each factor, and the red line representing the measured average concentrations. Tentative structures assigned to the individual ions are denoted in each figure.


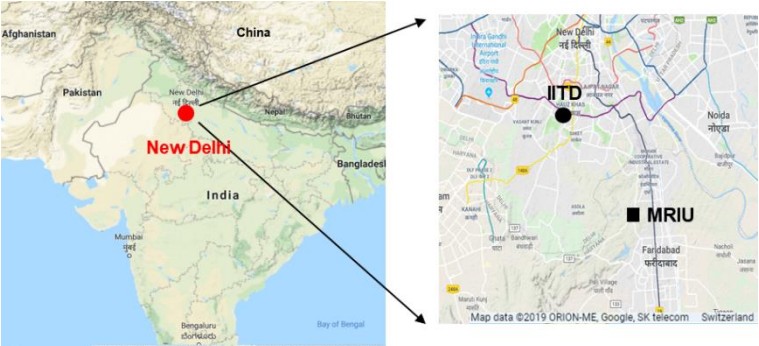


**Fig. 1** Maps of the study region and the two sampling sites (from Google Maps). The black circle denotes the
urban site at IITD and the black square the suburban site at MRIU.

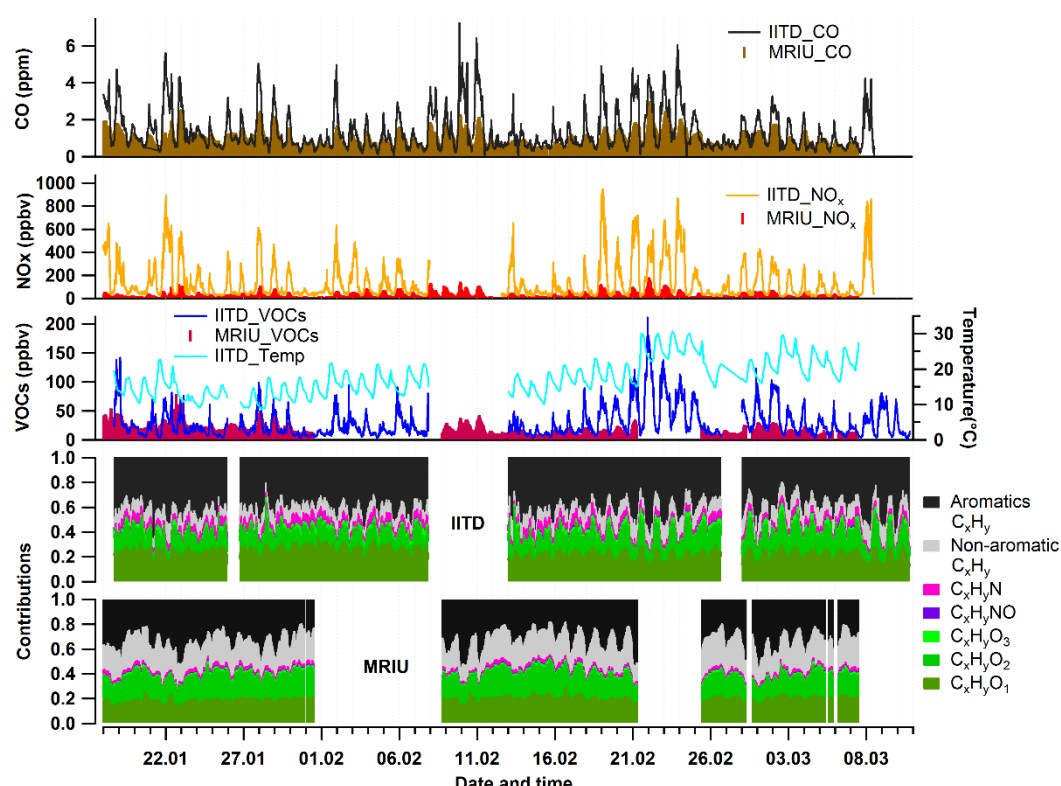


**Fig. 2** Temporal variations of CO, $NO_x$, analyzed VOC mixing ratios and temperature as well as the
contributions of seven VOC families at the two sites.

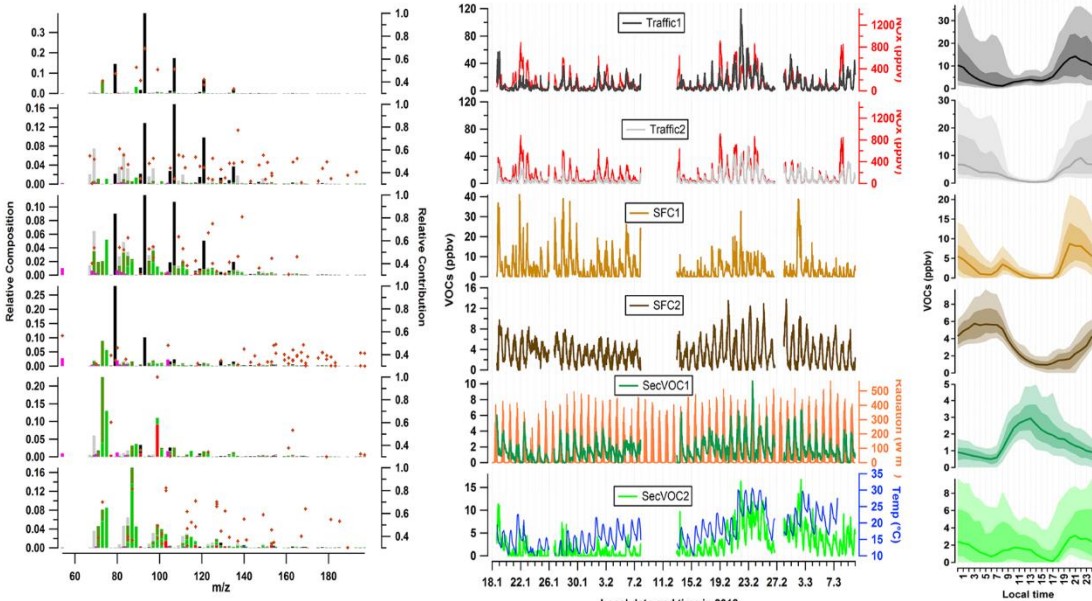


Fig. 3 PMF results at IITD, showing (a) factor profiles (b) time series and (c) diurnal patterns. In (a), the left axis for each factor profile is the relative composition of each factor (i.e., horizontal sum is 1) and the right axis is the relative contribution of each factor to a given ion. Ions are colored based on the seven family classes, as described in Section 3.1. (b) Temporal evolution of resolved factors at IITD, together with external reference data, e.g. NOx, CO, solar radiation and temperature. (c) Medians of diurnal cycles of factors at IITD, shaded with interquartile ranges as well as the 10[th] and 90[th] percentiles.

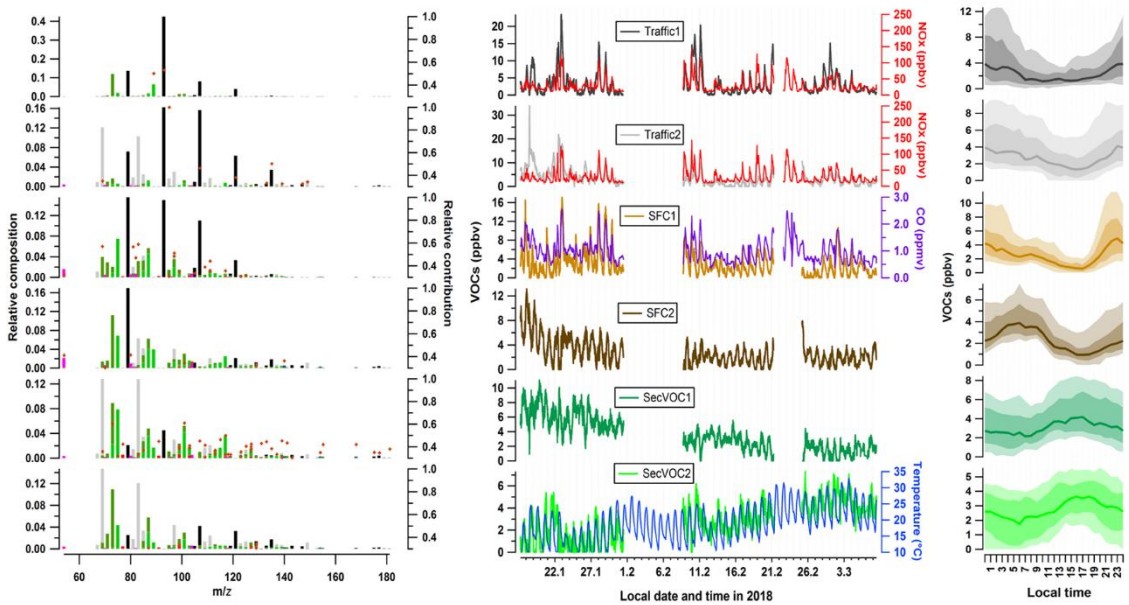

847

Fig. 4 PMF results at MRIU, showing (a) factor profiles (b) time series, and (c) diurnal patterns. (a) Relative composition (left axis) and relative contribution (right axis) of each factor to a given ion. Ions are colored based on the seven family classes, as described in Section 3.1. (b) Temporal evolution of resolved factors at MRIU, together with external reference data, e.g. NOx, CO, solar radiation and temperature. (c) Medians of diurnal

cycles of factors at MRIU, shaded with interquartile ranges as well as the 10th and 90th percentiles.

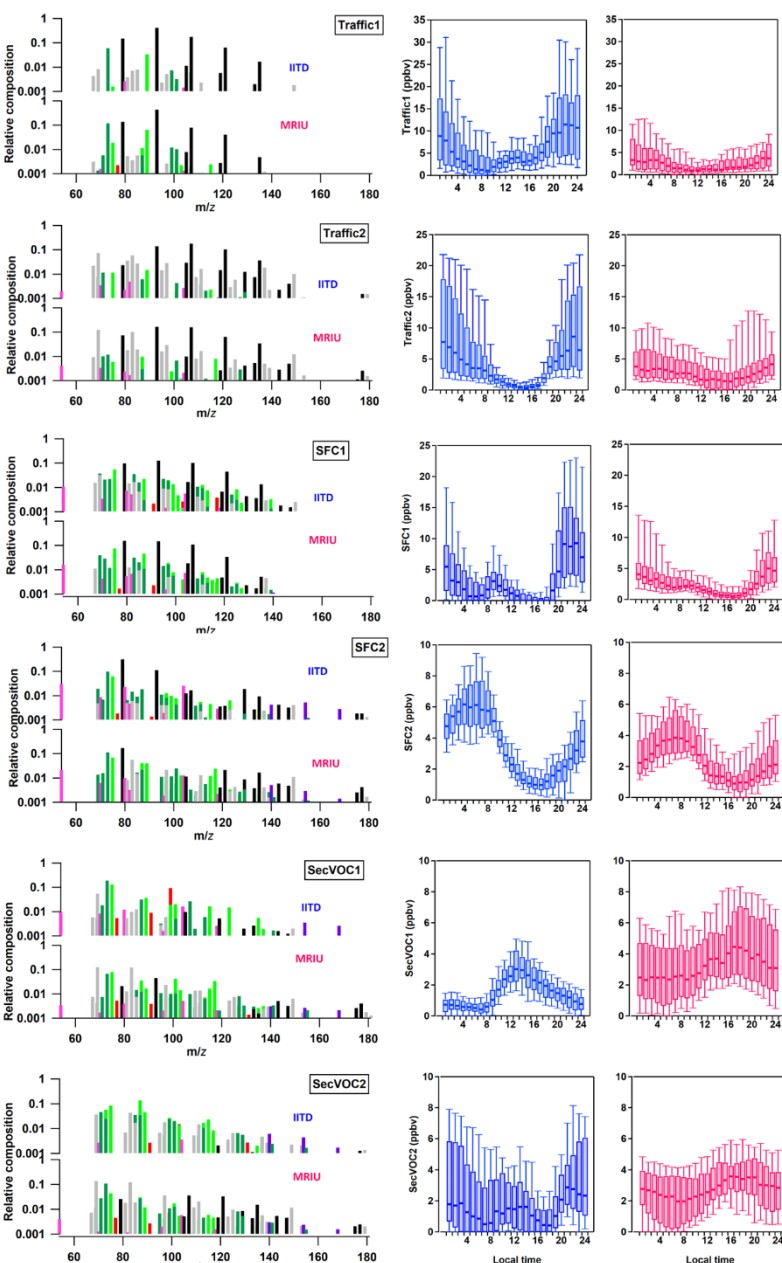


**Fig. 5** Comparisons of averaged factor diurnal patterns and factor profiles at the two sites. The left panels
present the factor profiles with the IITD spectrum on top and the MRIU spectrum on bottom, color coded by the
VOC families described in Section 3.1. The right panels show the box and whisker plots of diurnal cycles at the
two sites, with the blue panel representing IITD and the red panel MRIU.

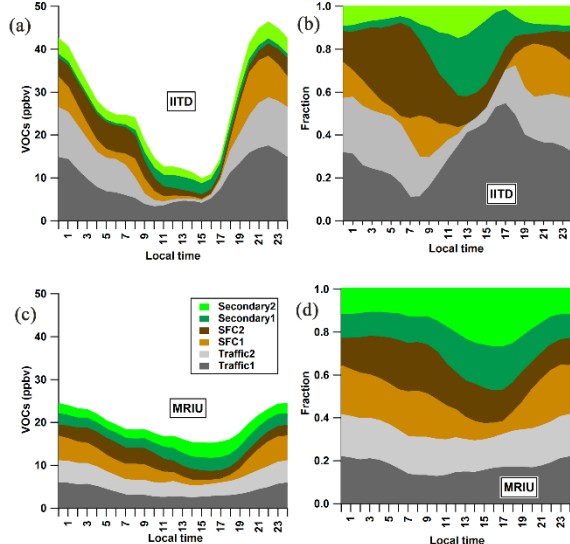


**Fig. 6** (a, c) Diurnal patterns of factor mixing ratios at the two sites, color-coded by the six retrieved factors. (b, d) Diurnal patterns of the fractional contributions of the factors at the two sites.

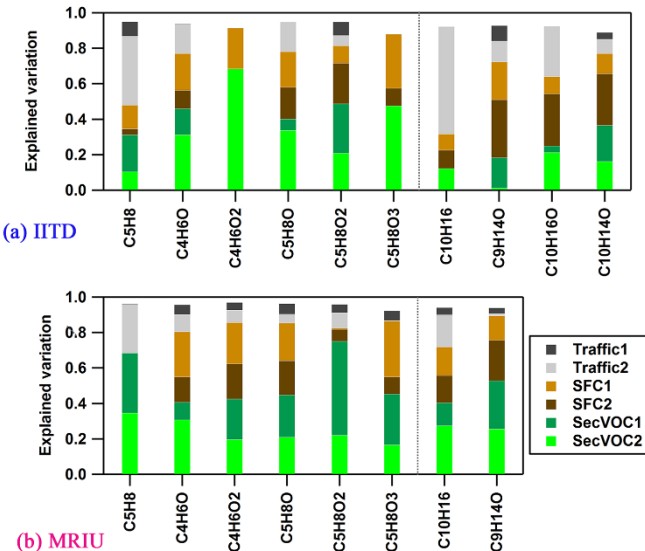


**Fig. 7** Explained variations of selected ions at the two sites, stacked such the sum is the total explained variation, color coded by the six factors (a) at IITD and (b) at MRIU. Missing ions at MRIU were excluded from PMF analysis due to low SNR. The possible candidates for these ions are isoprene ($C_5H_8$), MVK+MACR ($C_4H_6O$), 2,3-butanedione ($C_4H_6O_2$), 3-methyl-3-butene-2-one ($C_5H_8O$), methyl methacrylate ($C_5H_8O_2$), monoterpenes ($C_{10}H_{16}$), and camphor ($C_{10}H_{16}O$), respectively.

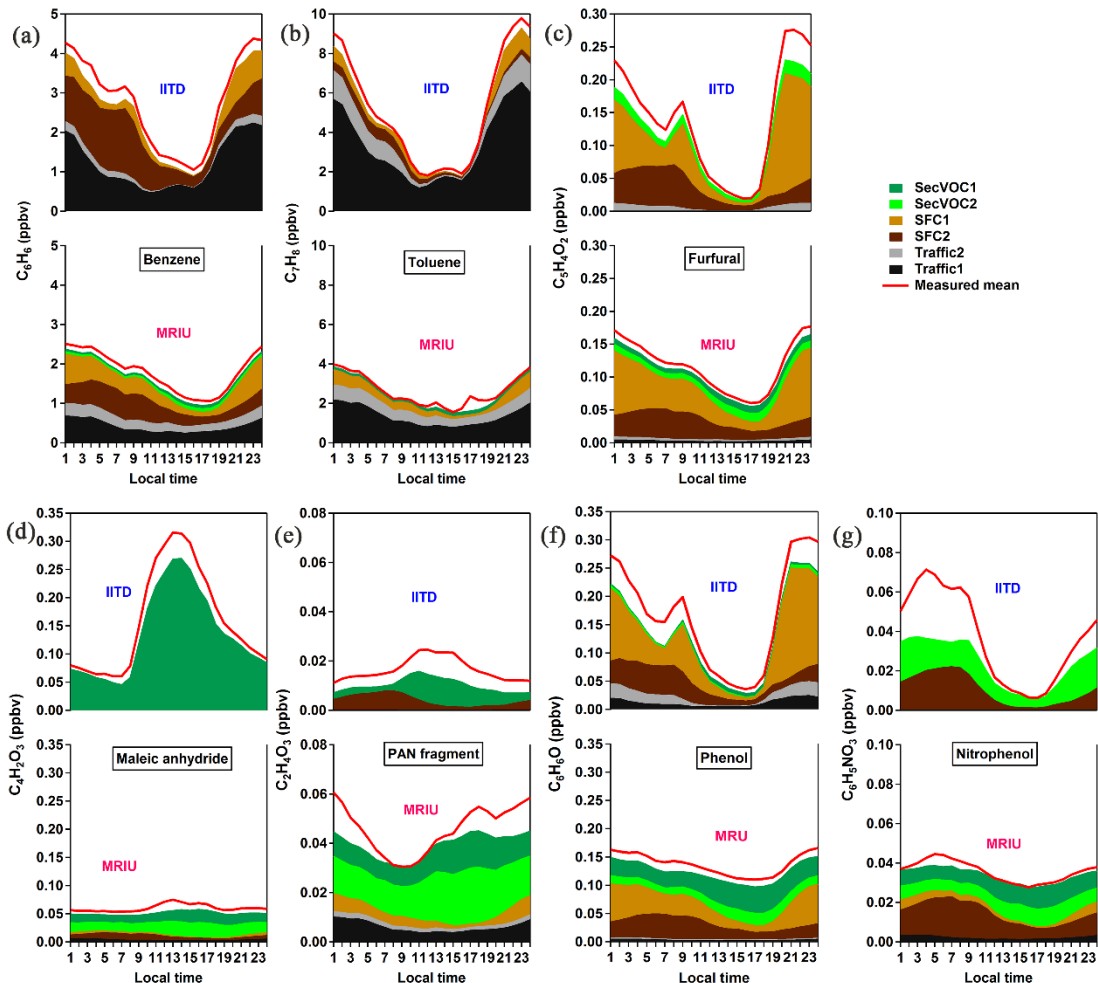


**Fig. 8** Diurnal cycles of selected marker ions, stacked with explained variations by each factor, and the red line
representing the measured average concentrations. Tentative structures assigned to the individual ions are
denoted in each figure.