# Peer review of "Source characterization of volatile organic compounds"

_Atmospheric Chemistry and Physics, 2020_

## Referee Comment (RC1) · Anonymous Referee #1 · 26 Mar 2020

The authors present measurements of volatile organic compounds (VOCs) using two proton transfer reaction time of flight mass spectrometers (PTR-ToF-MS) at an urban and suburban location in New Delhi in the wintertime. The positive matrix factorization (PMF) model is used to apportion various VOCs to different sources for each site. The authors find that six VOC factors could explain the observations for both sites, two factors related to traffic emissions, two to solid fuel combustion, and two secondary factors. The authors do a good job justifying the naming conventions for each factor at each site. However, it would be beneficial if these discussions were extended to back trajectory comparisons for the two sites, and comparisons to previous publications. Also, further discussion regarding VOCs excluded from the analysis (all masses above m/z 60) or not measured (alkanes, alkenes) that could dominate the VOCs mass, and

reactivity should be performed. This manuscript is suited for publication in ACP after the above revisions.

Specific comments

The authors could run a PMF combining the two datasets in one Matrix to generate common solutions. This way a possibly better statistical separation of the factors between the two sites could be achieved. Also, when comparing the two sites it would be great if a back-trajectory analysis could be performed in order to support that emissions from the city center travel from IITD to MRIU and therefore represent more aged air.

Line 143: How much of the total mass, and reactivity do these masses represent? Also, there are more m/z's not included in the mass list chosen for PMF like formic acid, ethanol, formaldehyde, and other masses related to fragments below m/z 60. These masses may represent the majority of the reactivity and should be further discussed, especially in the context of quantifying the effect of each emission source. I consider that results from the PMF when including the lower molecular weight compounds should at least be presented in detail in the SI and further discussed in the main text.

Line 164: The time that the temperature is higher from 21-25 Feb the VOCs are not measured at MRIU. Could the authors provide a box and whiskers of the temperature for the two periods? Also, for both periods, could they provide the temperature changes only when VOC measurements exist? This goes for Fig. S1 too. Could they provide the 25th and 75th percentile for Fig S1?

Line 180: What about comparing the absolute concentrations. Do these masses show the same diurnal patterns at the two sites? Please elaborate more.

Line 182-190: It will be more informative to include all the diurnal profiles for (i) the different VOC families, (ii) NOx, and (iii) CO for both sites in a supplementary figure.

Line 230: Here and throughout the text, it will be nice if the authors add for each chemical formula the actual m/z of detection in order for the readers to connect it better to the Figures.

Line 234: Please perform a more detail comparison to NOx. What is the R2 of (i) traffic1 vs NOx, (ii) traffic2 vs NOX, and (iii) traffic1+traffic2 vs NOx?

Line 237: this trend could also be driven by meteorology. A higher boundary layer leads to more dilution therefore a decrease of pollutants midday.

Line 238-243: The differences discussed between traffic1 and traffic2 are not supported by the figures. Traffic1 is in general higher than traffic2 even during nighttime when traffic2 is supposed to dominate the traffic emissions due to the heavy-duty vehicles. Also, the diurnal profiles of the two factors are very similar. The only difference is an increase of traffic1 midday that could also be due to the influence of other sources to traffic1, e.g. cooking. Please discuss more in the main text and provide more proof for the naming of these factors. Comparing to other studies of traffic emissions using PTR-ToF-MS could be of value, e.g. Gentner, et al. (1).

Line 243-245: This would be the dominant VOC source based on the VOCs included in the PMF analysis. It would be beneficial to include a discussion of the VOCs not included in the PMF analysis, especially smaller alkanes and alkenes that could substantially contribute to the total VOCs.

Line 246-270: The SFC factors can be further compared to previous studies in more detail. Sekimoto, et al. (2) found a high- and low-temperature factor related to biomass burning emissions. The factors found in this study are available to the public and could, therefore, be used to directly compare to this study.

Line 292-293: Define "similar", e.g. within XX% for XX% of the masses. Masses that are higher should be discussed.

Line 308-327: The discussion of the oxygenated factors and the comparison of the two

sites in this paragraph is hard to follow.

Line 264: Anthropogenic monoterpenes can also originate from fragranced volatile chemical product usage. Emissions from fragrances and personal care products have been found to coincide with traffic emissions (3, 4) which would explain why the monoterpenes are loaded in the traffic factor. Further discussion on this topic may be of interest.

Line 374-375: This sentence doesn't make sense.

Line 376-377: More detailed comparison of the results of this study to previous studies is required. For example, what is the ratio of the oxygenated compounds relative to aromatics for previous studies and this study?

Line 406-407: Give numbers. It is a 50-50 split between traffic and SFC emissions for the majority of the time.

Line 409: At MRIU it is again a 50-50 split. Precise discussion of the differences with numbers will be great in this paragraph.

Line 412-414: No discussion on toluene that is different than benzene is provided. Toluene is attributed predominantly to traffic at both sites. Wouldn't the authors expect toluene emissions from biomass burning based on existing literature cited in this manuscript? A detailed comparison of the ratios of aromatics to different studies may be of value here.

Line 463-466: A discussion on the missing VOCs should be performed if the comparison of the different emissions is made. What is the contribution in mass, and reactivity of the missing VOCs? How is it expected to distribute among the different emission sources based on inventories?

Comments on Figures Figure 5-6: Add error bars to the diurnal profiles. If it gets too busy split into two figures. A way to better focus on the differences in the factors from the two sites could be to plot the following: Traffic1(IITD) – Traffic1(MRIU) / Traffic1(MRIU) VS m/z This will better show the relative difference between the two. Now differences between many masses that are low in concentration but could still be important markers are not shown at all. Why is it that for Figure 6 the two oxygenated factors from both sites are not included? Why not extend figure 5 and add the two oxygenated factors there?

Figure 6: Label is missing the triangle markers explanation. It will also be better to add the SecVOC factors for both sites in this comparison.

Figure 8: Add suggested compounds in parenthesis.

Technical comments

Line 19: Explain abbreviation "IITD". Line 21: Explain abbreviation "MRIU". Line 36: Change to "gas-to-particle partitioning". Line 69: Change to "of the VOC pollution levels" Line 77: Maybe add the number of VOCs used for PMF since you already discussed what has been used in the literature before. Line 145: Define "extremely". Line 169: How would they have an impact on the emission profiles? Please elaborate. Line 205: Maybe rephrase. Sentence too long and hard to follow. Line 211: delete "was resolved". Also, define reasonable. What is the contribution of hydrocarbons in the secondary factors? Line 216: A graph providing the comparison of the different spectra in the SI will be more informative. Fig S4 is not informative regarding the comparison of the different a-value results. Line 291: delete "that". Also, why would the change in concentrations result in a better factor separation? Please, elaborate. This sentence is in general not easy to follow. Based on which figure are the authors concluding this? Line 247-248: What is the meaning of the chemical formulas written here? Line 293: Delete "very". Line 311: Delete "very" and define how different. Line 351: Provide m/z and chemical formula for the broader audience. Line 354: Change to "alcohols". Line 357: Delete "very". These values are low only compared to the primary anthropogenic emissions but a considerable value for biogenic emissions. Line 369: Delete "in". Line 385: Replace "by" with "but can". Line 399: Replace to "of the low".

Line 418: Please give a number for "majority". Line 419: replace "mainly" with "by XX% and XX% during the night and morning, respectively,". Line 446: Please support with back trajectory. Line 468-469: Please support with back trajectory.

References

1. Gentner DR, et al. (2013) Chemical Composition of Gas-Phase Organic Carbon Emissions from Motor Vehicles and Implications for Ozone Production. Environmental Science & Technology 47(20):11837-11848. 2. Sekimoto K, et al. (2018) High- and low-temperature pyrolysis profiles describe volatile organic compound emissions from western US wildfire fuels. Atmos. Chem. Phys. 18(13):9263-9281. 3. Coggon MM, et al. (2018) Diurnal Variability and Emission Pattern of Decamethylcyclopentasiloxane (D5) from the Application of Personal Care Products in Two North American Cities. Environmental Science & Technology 52(10):5610-5618. 4. McDonald BC, et al. (2018) Volatile chemical products emerging as the largest petrochemical source of urban organic emissions. Science 359(6377):760-764.

---

## Referee Comment (RC2) · Anonymous Referee #2 · 27 Mar 2020

The authors performed measurements of VOCs, NOx and CO at two locations in New Delhi, India: an urban and a suburban location. For VOC measurements, two proton transfer reaction time of flight mass spectrometers (PTR-ToF-MS) were used. VOC data was interpreted using positive matrix factorization (PMF). The authors find six factors explaining observations for both sites reasonable well: Traffic1, Traffic2, Solid-FuelCombustion1, SolidFuelCombustion2, Secondary1 and Secondary2. Overall, the manuscript is suited for publication in ACP after revisions.

Specific comments:

1) The authors explain that they excluded ions such as methanol, acetaldehyde, acetone, and acetic acid with extremely high mixing ratios. A comprehensive list would be useful; I assume it also includes Formaldehyde, (which is hard to quantify with PTR-

[Figure]

MS, but it would be worth mentioning). It would be useful to quantify all ions not taken into account, and plotting a diurnal cycle of these compounds. Although I'm not an expert in PMF, I wonder if downscaling these signals before feeding it to the PMF algorithm would solve the issue that these signals dominate PMF factors. More discussion is needed on that.

2) Diurnal patterns of Traffic1 and Traffic2 in Figures 3 and 4 are quite similar; Assigning Traffic2 to heavy-duty vehicles seems to be problematic, since the diurnal pattern of this factor is very close to zero (including its 90th percentile) from 10 am to 5 pm, which is exactly the time window where the ban of heavy-duty vehicles is lifted.

3) (Line 245+:) Are there numbers on how much traffic increased in New Delhi during the last two decades? Is traffic really the dominant VOC source when large VOC signals are not accounted for in PMF (acetone etc.)

4) Evaluation of biogenic VOCs: Line 359: please avoid "very low" and similar ill-defined expressions; I don't think that 0.46 ppbv is "very low"; Personal care products may be another possible source of Monoterpenes.

Technical comments:

Abstract: avoid abbreviations "IITD" and "MRIU" Line 44: no need for "natural" when talking about BVOCs Line 54: "high atmospheric reactivity and higher SOA yield" - need reference Line 64: "The critical air quality problems have left India with high death rates from ..." need reference or numbers Line 71: "..., a comprehensive investigation *of* VOC pollution levels..." Line 75" "..pointed out the lack of ..." Line 93: "The study site is approximately 150 m north of a busy street and surrounded by several streets as well" - unclear Line 97: "The site is located to the southeast of higher elevation terrain as shown in Figure 1" - unclear Line 116" use the term *reduced* electric field (E/N) Line 124: *Volume* mixing ratios Line 125: Please add a note of how and how often background measurements were performed Line 203: "... nor included in the PMF analysis ..." -> " ... or not included in this PMF analysis ..." Line 208: please

define "very high" Line 252: "These ions are tentatively attributed to phenolic compounds and furans" - reference needed Line 279: "However, many of these ions can be formed rapidly during daytime and may have a short lifetime owing to partitioning to the condensed phase and/or heterogeneous processes" - plese re-phrase: I think this is suggested by the data, not by the identidy of the compounds. Line 287: "Major fractions of alkyl nitrates (RONO2) are detected as (ROHÂůH+) fragment ions by the PTR-ToF-MS" - please cite Aoki et al: "Detection of C1–C5 alkyl nitrates by proton transfer reaction time-of-flight mass spectrometry" or similar Line 350: "But more importantly, differences in SecVOC *is probably owing to that chemical ...*" - please rephrase; hard to follow.

---

## Author Comment (AC1) · 5 Jun 2020

**Responses (text in blue) to comments by Referee #1 (text in black)**

We thank the referee for the valuable comments which have greatly helped us to improve the manuscript. Please find below our point-by-point responses (in blue) after the referee comments (in black). The changes in the revised manuscript are written *in italic*.

**Referee #1**

**General comments**

The authors present measurements of volatile organic compounds (VOCs) using two proton transfer reaction time of flight mass spectrometers (PTR-ToF-MS) at an urban and suburban location in New Delhi in the wintertime. The positive matrix factorization (PMF) model is used to apportion various VOCs to different sources for each site. The authors find that six VOC factors could explain the observations for both sites, two factors related to traffic emissions, two to solid fuel combustion, and two secondary factors. The authors do a good job justifying the naming conventions for each factor at each site. However, it would be beneficial if these discussions were extended to back trajectory comparisons for the two sites, and comparisons to previous publications. Also, further discussion regarding VOCs excluded from the analysis (all masses above m/z 60) or not measured (alkanes, alkenes) that could dominate the VOCs mass, and reactivity should be performed. This manuscript is suited for publication in ACP after the above revisions.

Reply: We thank the referee for the constructive comments. The discussion has been extended with back-trajectory analysis and external support from literature.

**Specific comments**

The authors could run a PMF combining the two datasets in one Matrix to generate common solutions. This way a possibly better statistical separation of the factors between the two sites could be achieved. Also, when comparing the two sites it would be great if a back-trajectory analysis could be performed in order to support that emissions from the city center travel from IITD to MRIU and therefore represent more aged air.

Reply: The referee raises two issues here: (1) run PMF combining the two datasets and (2) back-trajectory analysis for the two sites. We discuss these issues separately below.

(1) We believe that a source apportionment on the combined datasets would not be significantly helpful for this paper. Specifically, although the two instruments were operated in the same setting conditions, these could not ensure the performance of the two PTR-ToF-MS 8000 were the same all the time over the entire sampling period. Therefore, even though the combined dataset in principle could result in a better comparison, combining the two datasets could bring in more uncertainties into the error matrix. Further, the VOC fingerprints might be rather different in different locations, for example, different types of rather local biomass burning are present. Therefore, the separation of the dataset allows for more locally adapted fingerprints.

(2) We agree that back-trajectory analysis improves the discussion on the transport of emissions from IITD to MRIU. This has been added to the manuscript, as discussed in lines *328-332* here and in response to specific comments (Specific comments Line 308-327, Technical comments Line 446 and Line 468-469) below.

*Line 331-335. Besides, Fig. S8 presents concentration weighted trajectory (CWT) plots of the six factors at both sites. Details of the back-trajectory and CWT analysis are shown in the Supplement. As shown in Fig. S8, high concentrations of primary factors are found both north and northwest of IITD, whereas high primary factors originate from southeast of MRIU, different from the direction of IITD.*

Line 143: How much of the total mass and reactivity do these masses represent? Also, there are more m/z's not included in the mass list chosen for PMF like formic acid, ethanol, formaldehyde, and other masses related to fragments below m/z 60. These masses may represent the majority of the reactivity and should be further discussed, especially in the context of quantifying the effect of each emission source. I consider that results from

the PMF when including the lower molecular weight compounds should at least be presented in detail in the SI and further discussed in the main text.

Reply: We agree with the referee that mass peaks below *m/z* 60 are important and interesting in terms of their reactivity, emission ratios and sources. These topics are being prepared for publication in a separate manuscript. (Tripathi et al., in preparation). In that paper the relative contributions from anthropogenic and biogenic sources of these ions are discussed. In addition, the discussion on mass contributions are combined in response to Specific comments, Line 243-245 #1 by Referee 1 below

Line 164: The time that the temperature is higher from 21-25 Feb the VOCs are not measured at MRIU. Could the authors provide a box and whiskers of the temperature for the two periods? Also, for both periods, could they provide the temperature changes only when VOC measurements exist? This goes for Fig. S1 too. Could they provide the 25th and 75th percentile for Fig S1?

Reply: We have replaced Fig. S1 with box-whisker plots showing temperature, separated by site and cold/warm periods. The comparison of temperature change during different periods, and at each site is based on the periods when VOC measurements exist.

*Line 173-177. Fig S1 shows the box-whisker plots of temperature during the two periods at the two sites. As shown in Fig. S1, the average temperature was 17 ºC during the cold period and 23 ºC during the warm period at IITD. At MRIU, the average temperature was very similar when VOCs were measured, with the cold days averaging at 16 ºC and the warm periods at 23 ºC. During the warm period, the mean temperature reached the minimum at 6:00 LT, 1 hour earlier than in the cold period.*

Line 180: What about comparing the absolute concentrations. Do these masses show the same diurnal patterns at the two sites? Please elaborate more.

Reply: We added the absolute concentrations.

*Line 189-194. At MRIU, the VOC family composition was similar, with the exception of a higher fraction and concentration of non-aromatic CxHy (23.2%, 4.2 ppbv), which was dominated by high C5H8 and C6H10 in the daytime compared to that at IITD. Besides, the averaged mixing ratios of the majority of the families were lower at MRIU than at IITD, except for non-aromatic CxHy and CxHyO2. The averaged mixing ratios of CxHyO1, CxHyO2, CxHyN, CxHyO3, and CxHyNO were 3.6 ppbv, 3.4 ppbv, 0.5 ppbv, 0.2 ppbv, and 0.09 ppbv, respectively.*

Line 182-190: It will be more informative to include all the diurnal profiles for (i) the different VOC families, (ii) NOx, and (iii) CO for both sites in a supplementary figure.

Reply: The requested diurnal profiles have been added to Fig. S2. Consequently, we added some discussion on the diurnal patterns of VOC families at both sites. The corresponding paragraph is rephrased.

*Line 197-209. Fig. S2 shows the diurnal patterns of $NO_x$, CO, and different VOC families at both sites. The nocturnal mixing ratios of most of the VOC families, as well as those of CO and $NO_x$ were higher than during daytime, with much greater diurnal variation at IITD than at MRIU. The spatial difference in the diurnal variation may be due to a lower influence of local emissions at the suburban MRIU site because of lower population density and fuel consumption. In addition, the relative proportions of the VOC families varied over time, indicating different emission patterns and oxidation chemistry. For instance, substantial contributions and concentrations of aromatic compounds and $C_xH_yNO_1$ were observed at night at IITD, indicating strong anthropogenic emissions in the urban area, such as traffic-related emissions and solid fuel combustion. Meanwhile, higher daytime contributions were found for the $C_xH_yO_1$ and $C_xH_yO_2$ families; and in particular the $C_xH_yO_3$ family peaked around midday at IITD, indicating tropospheric aging and secondary formation during daytime. Moreover, the nocturnal concentrations of all the gas phase species were higher at IITD compared to*

*MRIU. During daytime, however, the concentrations of non-aromatic $C_xH_y$, $C_xH_yO_2$, as well as CO at MRIU were higher compared to those at IITD.*

Line 230: Here and throughout the text, it will be nice if the authors add for each chemical formula the actual m/z of detection in order for the readers to connect it better to the Figures.

Reply: We have added the *m/z* values. Besides, the actual *m/z* values of all the ions included in PMF are shown in Table S1 and Table S2.

Line 234: Please perform a more detail comparison to NOx. What is the R2 of (i) traffic1 vs NOx, (ii) traffic2 vs $NO_x$, and (iii) traffic1+traffic2 vs NOx?

Reply: We now address this as follows in the manuscript:

*Line 257-258. The correlation of Traffic 2 with $NO_x$ ($R^2=0.76$) is better than either Traffic1 with $NO_x$ ($R^2=0.55$) or their sum (Traffic1+Traffic2) with $NO_x$ ($R^2=0.67$).*

Line 237: this trend could also be driven by meteorology. A higher boundary layer leads to more dilution therefore a decrease of pollutants midday.

Reply: We agree that the boundary layer could lead to dilution of pollutants in the midday and accumulation during night. However, PBL dilution will affect most pollutants similarly (except for long-lived species, which may be equally abundant in the upper layer), i.e. maintaining factor-to-factor ratios. This is not observed for Traffic1 and Traffic2, for which both the time of initial decrease and the relative rate of decrease differ. Therefore, we interpret the differences in the factor diurnals to represent real differences in the traffic emissions patterns.

Line 238-243: The differences discussed between traffic1 and traffic2 are not supported by the figures. Traffic1 is in general higher than traffic2 even during nighttime when traffic2 is supposed to dominate the traffic emissions due to the heavy-duty vehicles. Also, the diurnal profiles of the two factors are very similar. The only difference is an increase of traffic1 midday that could also be due to the influence of other sources to traffic1, e.g. cooking. Please discuss more in the main text and provide more proof for the naming of these factors. Comparing to other studies of traffic emissions using PTR-ToF-MS could be of value, e.g. Gentner, et al. (1).

Reply: We added some discussion clarifying the differences between Traffic1 and Traffic2. As shown by Platt et al. (2017), cold start from gasoline vehicles emit high amounts of VOCs while heavy-duty vehicles contribute much less to VOCs but might be dominating contributor to BC and NOx. Therefore, the heavy-duty vehicles will still not dominate traffic emissions. Besides, the compounds listed in the suggested reference paper (Gentner et al., 2013), however, are focusing on ions very different from this study. Many of the high concentration species are not detectable and/or poorly quantified by PTR-ToF-MS. Therefore, a few aromatic ions are selected for comparison.

*Line 261-270. As shown in Fig. S7, the ratio of Traffic 2 to Traffic 1 is very low during the day time and starts to increase slightly from 16:00 LT. Although both Traffic1 and Traffic2 are high during the night with their maximum concentrations around 21:00 LT, the Traffic2/Traffic1 ratio is as low as 0.6 at that time. The ratio increases overnight with a sharp increase during the early morning, reaching a maximum value of 1.4 at 07:00 LT, suggesting that Traffic1 is the dominant traffic source both day and night. This can be explained as cold start emissions from gasoline vehicles emit high amounts of VOCs while heavy-duty vehicles contribute much less to VOCs but more to BC and $NO_x$ (Platt et al., 2017). Further, the spectrum of Traffic2 is characterized by high fractions of high mass aromatic compounds. For instance, the ratio of C8/C7 aromatics is tripled in Traffic2 compared to that in Traffic1, which is similar to the ratio of the emission factors from previous studies (3.5 times (Gentner et al., 2013)).*

Line 243-245: This would be the dominant VOC source based on the VOCs included in the PMF analysis. It would be beneficial to include a discussion of the VOCs not included in the PMF analysis, especially smaller alkanes and alkenes that could substantially contribute to the total VOCs.

Reply: We have added some discussion in Section 3.3.

*Line 393-401. In addition, several ions below $m/z$ 60 that can be detected by the PTR-ToF-MS are excluded from PMF as discussed in Sec. 2.3, such as methanol, acetaldehyde, acetone and acetic acids. These ions are 3-4 times higher than the dominant ions in the PMF analysis, possibly owing to much higher emission rates and natural abundance. Besides, other excluded compounds such as $C_1$-$C_4$ alkanes and $C_1$-$C_4$ alkenes which are not detectable by the PTR-ToF-MS are substantial contributors to the total VOC mixing ratio as well. However, these ions are minor contributors to SOA formation and only substantially contribute to the formation of ozone, which is a major issue in summer. Although the mixing ratio of the sum of VOCs in the PMF only accounts for 39.6 % at IITD and 24.2 % at MRIU (Fig. S11), many of these compounds are the dominant precursors in terms of SOA formation (Wu and Xie, 2017, 2018).*

Line 246-270: The SFC factors can be further compared to previous studies in more detail. Sekimoto, et al. (2) found a high- and low-temperature factor related to biomass burning emissions. The factors found in this study are available to the public and could, therefore, be used to directly compare to this study.

Reply: Based on the diurnal patterns, we expect that the SFC factors in this study are largely associated with domestic heating. The suggested reference paper (Sekimoto et al., 2018), however, focused on open burning of wildfires. These two are substantially different types of combustion and thus we believe these profiles are not suitable for comparison to this study. Here, we compare the emission ratios reported by Bruns et al. (2017) of residential wood burning.

*Line 278-281. The relative fractions of emission factors (EF, in mg per kg fuel) of $C_5H_4O_2$ (100 mg kg$^{-1}$), $C_6H_6O$ (110 mg kg$^{-1}$), and $C_6H_6O_2$ (60 100 mg kg$^{-1}$) are similar to their relative concentrations in SFC1 (i.e. $C_5H_4O_2$ ($m/z$ 97.028, 0.019), $C_6H_6O$ ($m/z$ 95.049, 0.021), and $C_6H_6O_2$ ($m/z$ 111.044, 0.012)), consistent with the identification of the factor.*

Line 292-293: Define "similar", e.g. within XX% for XX% of the masses. Masses that are higher should be discussed.

Reply: Done. We have added some discussion.

*Line 326-327. While the factor profiles are similar for the primary factors (i.e. Traffic1, Traffic 2 and SFC1 at both sites share the same major ions) their diurnal patterns are very different.*

Line 308-327: The discussion of the oxygenated factors and the comparison of the two sites in this paragraph is hard to follow.

Reply: We rephrased the paragraph and included back-trajectories to support the statements.

*Line 351-353. Further, the IITD SecVOC1 shows additional origins apart from the directions of primary factors, indicating influences from transport (Fig. S8). The SecVOC2 at IITD, however, is comparable to the MRIU SecVOC1 in terms of factor fingerprint (Fig. S9).*

*Line 360-362. Besides, as shown in Fig. S8, SecVOC2 at IITD and SecVOC1 at MRIU exhibit similar geographical origins as the primary factors at the respective sites, suggesting local oxidation of primary emissions.*

*Line 366-368. As shown in Fig. S8, high concentrations of MRIU SecVOC2 originate mainly from north and northwest directions, consistent with the location of the IITD site. Therefore, it is possible that MRIU SecVOC2 represents oxidized VOCs on a relatively regional scale.*

Line 264: Anthropogenic monoterpenes can also originate from fragranced volatile chemical product usage. Emissions from fragrances and personal care products have been found to coincide with traffic emissions (3, 4) which would explain why the monoterpenes are loaded in the traffic factor. Further discussion on this topic may be of interest.

Done. We added some discussion on anthropogenic monoterpenes.

*Line 418-420. One recent paper showed that fragrances and personal care products may be an important emission source of urban monoterpenes and are correlated with traffic emissions (McDonald et al., 2018).*

Line 374-375: This sentence doesn't make sense.

Reply: This sentence was attempting to summarize Fig 8 and the discussion in the paragraph. Since all the information is given anyway and the sentence is misleading, we deleted it.

Line 376-377: More detailed comparison of the results of this study to previous studies is required. For example, what is the ratio of the oxygenated compounds relative to aromatics for previous studies and this study?

Reply: This sentence was attempting to show that SFC is one possible source of these ions. Besides, the OVOCs/benzene ratios as suggested are not suitable for comparing directly to the explained variation in this study. We have revised the sentence and used a comparison of emission factors.

*Line 429-432. However, at both sites, considerable amounts of these ions are also explained by emissions from SFC. Previous studies showed that the emission factors (EF, in mg per kg fuel) of $C_4H_6O_{1-2}$ are comparable to that of benzene (200 mg $kg^{-1}$), and the EFs for $C_5H_8O_{1-2}$ and $C_{10}H_{16}O$ are comparable to that of toluene (27 mg $kg^{-1}$) in biomass/wood burning emissions (Bruns et al., 2017; Koss et al., 2018).*

Line 406-407: Give numbers. It is a 50-50 split between traffic and SFC emissions for the majority of the time.

Reply: We added some discussion on the ratios of traffic and SFC emissions.

*Line 461-464. As shown in Fig. 8 (a), at IITD benzene originates to a significant extent from traffic emissions (53% on average) over the whole day and in addition from the SFC factors (47% on average), largely at night. The traffic fraction is lowest (29%) in the early morning and increases during daytime, with a maximum of 74 % around 17:00-18:00 LT, with the opposite trend for the SFC fraction.*

Line 409: At MRIU it is again a 50-50 split. Precise discussion of the differences with numbers will be great in this paragraph.

Reply: We added some discussion.

*Line 469-470. Although the traffic fraction increases during daytime at both sites, it explains a maximum of 74% at IITD and 49% at MRIU around 17:00-18:00 LT, with the opposite behavior for the SFC fraction.*

Line 412-414: No discussion on toluene that is different than benzene is provided. Toluene is attributed predominantly to traffic at both sites. Wouldn't the authors expect toluene emissions from biomass burning based on existing literature cited in this manuscript? A detailed comparison of the ratios of aromatics to different studies may be of value here.

Reply: We added some discussion on Toluene.

*Line 472-479. The traffic fraction is as high as 90 % around 18:00 LT and reaches a minimum of 67 % around 10:00 LT at IITD, which is much higher than that of benzene. The benzene/toluene (B/T) of the traffic factor ranges from 0.34 to 0.40, which is comparable to the emission factors of benzene/toluene for gasoline emissions (0.58, Gentner et al., 2013). Besides, the traffic fraction is lowest in the morning, which is associated with the strong emissions of SFC in the morning and high traffic emissions during the rush hour in the late afternoon.*

*The B/T is 3.4 for the SFC factors, which is in the reported range of 2-7 for residential biomass/wood burning (Bruns et al., 2017; Koss et al., 2018). Similarly, traffic is also the highest source of toluene at MRIU, ranging from 67%-77%.*

Line 463-466: A discussion on the missing VOCs should be performed if the comparison of the different emissions is made. What is the contribution in mass, and reactivity of the missing VOCs? How is it expected to distribute among the different emission sources based on inventories?

Reply: This point was discussed in detail above (Line 143). Simply put, these topics are interesting, but are the focus of another paper prepared by our collaborators.

**Comments on Figures**

Figure 5-6: Add error bars to the diurnal profiles. If it gets too busy split into two figures. A way to better focus on the differences in the factors from the two sites could be to plot the following: Traffic1(IITD) – Traffic1(MRIU) / Traffic1(MRIU) VS m/z This will better show the relative difference between the two. Now differences between many masses that are low in concentration but could still be important markers are not shown at all. Why is it that for Figure 6 the two oxygenated factors from both sites are not included? Why not extend figure 5 and add the two oxygenated factors there?

Figure 6: Label is missing the triangle markers explanation. It will also be better to add the SecVOC factors for both sites in this comparison.

Reply: We revised the axes in log scale and combined all the six factors into Figure 5.

Figure 8: Add suggested compounds in parenthesis.

Reply: Done.

**Technical comments**

Line 19: Explain abbreviation "IITD". Line 21: Explain abbreviation "MRIU".

Reply: We removed these abbreviations from the abstract.

Line 36: Change to "gas-to-particle partitioning".

Reply: Done. We revised the sentence.

*Line 39-40. The chemical transformation of VOCs forms less-volatile compounds and can contribute to gas-to-particle partitioning either by new particle formation or condensation on existing particles.*

Line 69: Change to "of the VOC pollution levels"

Reply: Done

Line 77: Maybe add the number of VOCs used for PMF since you already discussed what has been used in the literature before.

Reply: Done.

*Line 83. The level, composition and source characteristics of different VOCs (158 ions at IITD and 90 ions at MRIU) were analyzed*

Line 145: Define "extremely".

Reply: Done.

*Line 150-151. 3-4 times higher than other ions were excluded from the input*

Line 169: How would they have an impact on the emission profiles? Please elaborate.

Reply: We cited Sekimoto et al. (2018) as an example, who illustrate the significant differences of profile and reactivity at high and low temperature.

Line 205: Maybe rephrase. Sentence too long and hard to follow.

Reply: Done.

*Line 225-226: At IITD, the concentrations of several aromatics were very high, being 10-20 times higher than those of the major phenols, and over 100 times higher than for the compounds with the lowest concentration.*

Line 211: delete "was resolved". Also, define reasonable. What is the contribution of hydrocarbons in the secondary factors?

Reply: We deleted reasonable.

Line 216: A graph providing the comparison of the different spectra in the SI will be more informative. Fig S4 is not informative regarding the comparison of the different a-value results.

Reply: We revised Fig. S4 (now Fig. S5).

*Line 239: and the result with an a-value=0.3 is presented in Figure S5*

Line 291: delete "that". Also, why would the change in concentrations result in a better factor separation? Please, elaborate. This sentence is in general not easy to follow. Based on which figure are the authors concluding this?

Reply: Compared to the ratios at MRIU, the IITD aromatics are an order of magnitude higher than other ions. Therefore, even though only a small fraction of aromatic compounds is apportioned to one factor, it will be high in relative concentration. This has been discussed in the paper. Besides, the higher concentrations could result in a higher contribution to Q for the same unexplained signal fraction. Thus, at IITD, the PMF will tend to push the signals into the non-aromatic factors to well explain the aromatics.

Line 247-248: What is the meaning of the chemical formulas written here?

Reply: For example, $C_6H_6(C_nH_{2n})O_1$ means $C_6H_6O$ (n=0), $C_7H_8O$ (n=1), $C_8H_{10}O$ (n=2), etc.

Line 293: Delete "very".

Reply: Done.

Line 311: Delete "very" and define how different.

Reply: Done. The paragraph has been revised according to Specific comments, Line 308-327.

Line 351: Provide m/z and chemical formula for the broader audience.

Reply: Done.

Line 354: Change to "alcohols".

Reply: Done.

Line 357: Delete "very". These values are low only compared to the primary anthropogenic emissions but a considerable value for biogenic emissions.

Reply: Done.

Line 369: Delete "in".

Reply: Done.

Line 385: Replace "by" with "but can".

Reply: Done.

Line 399: Replace to "of the low".

Reply: Done.

Line 418: Please give a number for "majority".

Reply: Done.

*Line 469-470. Although the traffic fraction increases during daytime at both sites, it explains a maximum of 74% at IITD and 49% at MRIU around 17:00-18:00 LT. with the opposite behavior of the SFC fraction.*

Line 419: replace "mainly" with "by XX% and XX% during the night and morning, respectively,".

Reply: Done

[revised manuscript text omitted]

---

## Author Comment (AC2) · 5 Jun 2020

**Responses (text in blue) to comments by Referee #2 (text in black)**

We thank the referee for the valuable comments, which have greatly helped us to improve the manuscript. Please find below our point-by-point responses (in blue) after the referee comments (in black). The changes in the revised manuscript are written *in italic*.

**Referee #2**

The authors performed measurements of VOCs, NOx and CO at two locations in New Delhi, India: an urban and a suburban location. For VOC measurements, two proton transfer reaction time of flight mass spectrometers (PTR-ToF-MS) were used. VOC data was interpreted using positive matrix factorization (PMF). The authors find six factors explaining observations for both sites reasonable well: Traffic1, Traffic2, Solid Fuel Combustion1, Solid Fuel Combustion2, Secondary1 and Secondary2. Overall, the manuscript is suited for publication in ACP after revisions.

**Specific comments:**

1) The authors explain that they excluded ions such as methanol, acetaldehyde, acetone, and acetic acid with extremely high mixing ratios. A comprehensive list would be useful; I assume it also includes Formaldehyde, (which is hard to quantify with PTR-MS, but it would be worth mentioning). It would be useful to quantify all ions not taken into account, and plotting a diurnal cycle of these compounds. Although I'm not an expert in PMF, I wonder if downscaling these signals before feeding it to the PMF algorithm would solve the issue that these signals dominate PMF factors. More discussion is needed on that.

Reply: We thank the referee for pointing out these interesting topics. The referee raises two issues here: (1) characteristics of excluded compounds and (2) application of downscaling technique. We discuss these issues separately below.

(1) This point was discussed in detail above (Line 143) of the reply to comments #1 by Referee 1 (Line 143), and our response is repeated here:

Reply: We agree with the referee that mass peaks below m/z 60 are important and interesting in terms of their reactivity, emission ratios and sources. These topics are being prepared for publication in a separate manuscript. (Tripathi et al., in preparation). In that paper the relative contributions from anthropogenic and biogenic sources of these ions are discussed. In addition, the discussion on mass contributions are combined in response to Specific comments, Line 243-245 #1 by Referee 1 below.

Line 393-401. In addition, several ions below m/z 60 that can be detected by the PTR-ToF-MS are excluded from PMF as discussed in Sec. 2.3, such as methanol, acetaldehyde, acetone and acetic acids. These ions are 3-4 times higher than the dominant ions in the PMF analysis, possibly owing to much higher emission rates and natural abundance. Besides, other excluded compounds such as  $C_1$ - $C_4$  alkanes and  $C_1$ - $C_4$  alkenes which are not detectable by the PTR-ToF-MS are substantial contributors to the total VOC mixing ratio as well. However, these ions are minor contributors to SOA formation and only substantially contribute to the formation of ozone, which is a major issue in summer. Although the mixing ratio of the sum of VOCs in the PMF only accounts for 39.6 % at IITD and 24.2 % at MRIU (Fig. S11), many of these compounds are the dominant precursors in terms of SOA formation (Wu and Xie, 2017, 2018).

(2) Although downscaling could in theory push the high concentration signals into other factors, in this study, we believe that applying this technique would bring more uncertainties. For instance, chosen of the downscaling parameter is critical. Therefore, without any reference/technical support, an unideal parameter could bring more uncertainties to both the source composition and contribution. Further, the VOC fingerprints might be rather different in different locations, for example, different types of rather local biomass burning are present. Therefore, locally adapted parameters might be needed for each dataset and each compound.

2) Diurnal patterns of Traffic1 and Traffic2 in Figures 3 and 4 are quite similar; Assigning Traffic2 to heavyduty vehicles seems to be problematic, since the diurnal pattern of this factor is very close to zero (including its 90th percentile) from 10 am to 5 pm, which is exactly the time window where the ban of heavy-duty vehicles is lifted.

Reply: We apologize for the mistake; heavy-duty vehicles are banned from 7:00-21:00 LT. The discrimination of Traffic1 and Traffic2 are revised in response to comments #1 by Referee 1 (Line238-243), and our response is repeated here.

Line 261-270. As shown in Fig. S7, the ratio of Traffic 2 to Traffic 1 is very low during the day time and starts to increase slightly from 16:00 LT. Although both Traffic1 and Traffic2 are high during the night with their maximum concentrations around 21:00 LT, the Traffic2/Traffic1 ratio is as low as 0.6 at that time. The ratio increases overnight with a sharp increase during the early morning, reaching a maximum value of 1.4 at 07:00 LT, suggesting that Traffic1 is the dominant traffic source both day and night. This can be explained as cold start emissions from gasoline vehicles emit high amounts of VOCs while heavy-duty vehicles contribute much less to VOCs but more to BC and  $NO_x$  (Platt et al., 2017). Further, the spectrum of Traffic2 is characterized by high fractions of high mass aromatic compounds. For instance, the ratio of C8/C7 aromatics is tripled in Traffic2 compared to that in Traffic1, which is similar to the ratio of the emission factors from previous studies (3.5 times (Gentner et al., 2013)).

3) (Line 245+:) Are there numbers on how much traffic increased in New Delhi during the last two decades? Is traffic really the dominant VOC source when large VOC signals are not accounted for in PMF (acetone etc.)

Reply: We have added the discussion on traffic increase.

Line 271-272. the registered vehicular population has tripled since 1994 and has reached 7.6 million.

We added discussion regarding the concentration of excluded VOCs replying to comments #1 by Referee 1 (Line243-245), and our response is repeated here.

Line 393-401. In addition, several ions below m/z 60 that can be detected by the PTR-ToF-MS are excluded from PMF as discussed in Sec. 2.3, such as methanol, acetaldehyde, acetone and acetic acids. These ions are 3-4 times higher than the dominant ions in the PMF analysis, possibly owing to much higher emission rates and natural abundance. Besides, other excluded compounds such as  $C_1$ - $C_4$  alkanes and  $C_1$ - $C_4$  alkenes which are not detectable by the PTR-ToF-MS are substantial contributors to the total VOC mixing ratio as well. However, these ions are minor contributors to SOA formation and only substantially contribute to the formation of ozone, which is a major issue in summer. Although the mixing ratio of the sum of VOCs in the PMF only accounts for 39.6 % at IITD and 24.2 % at MRIU (Fig. S11), many of these compounds are the dominant precursors in terms of SOA formation (Wu and Xie, 2017, 2018).

4) Evaluation of biogenic VOCs: Line 359: please avoid "very low" and similar ill-defined expressions; I don't think that 0.46 ppbv is "very low"; Personal care products may be another possible source of Monoterpenes.

Reply: We agree 0.46 ppbv is not "very" low in terms of BVOCs. We modified the text and added the discussion on personal care products in response to comments #1 by Referee 1 (Line 264), and our response is repeated here.

Line 418-420. One recent paper showed that fragrances and personal care products may be an important emission source of urban monoterpenes and are correlated with traffic emissions (McDonald et al., 2018).

**Technical comments:**

Abstract: avoid abbreviations "IITD" and "MRIU"

Reply: We adjusted the abstract replacing the abbreviations of IITD with *the urban site* and MRIU with *the suburban site*.

Line 44: no need for "natural" when talking about BVOCs

Reply: Done. We removed "natural"

Line 54: "high atmospheric reactivity and higher SOA yield" - need reference

Reply: Done.

Line 64: "The critical air quality problems have left India with high death rates from ..." need reference or numbers

Reply: Done

Line 71: "..., a comprehensive investigation \*of\* VOC pollution levels..."

Reply: Done

Line 75" "...pointed out the lack of ..."

Reply: Done.

Line 93: "The study site is approximately 150 m north of a busy street and surrounded by several streets as well" - unclear

Reply: We revised the sentence.

*Line 95-96. The study site is approximately 80 m north of a busy street and is surrounded by several streets inside the campus as well.*

Line 97: "The site is located to the southeast of higher elevation terrain as shown in Figure 1" - unclear

Reply: We revised the sentence.

Line 101-102. Besides, the northeast territory of MRIU is of slightly higher elevation compared to the sampling site as shown in Fig. 1.

Line 116" use the term \*reduced\* electric field (E/N)

Reply: Done.

Line 124: \*Volume\* mixing ratios

Reply: Done.

Line 125: Please add a note of how and how often background measurements were performed

Reply: We have added the background performance.

Line 124-125. The background measurements were performed using a dry zero air cylinder every two weeks

Line 203: "... nor included in the PMF analysis ..." -> " ... or not included in this PMF analysis ..."

Reply: Done. We revised it.

Line 208: please define "very high"

Reply: We revised the sentence.

*Line* 225-226: *At IITD, the concentrations of several aromatics were very high, being* 10-20 *times higher than those of the major phenols, and over 100 times higher than for the compounds with the lowest concentration.*

Line 252: "These ions are tentatively attributed to phenolic compounds and furans" - reference needed

**Reply: Done.**

Line 279: "However, many of these ions can be formed rapidly during daytime and may have a short lifetime owing to partitioning to the condensed phase and/or heterogeneous processes" - please re-phrase: I think this is suggested by the data, not by the identity of the compounds.

**Reply: We revised the sentence.**

Line 309-313. The time series of SecVOC1 follows that of the solar radiation, which has a regular contribution cycle during daytime. The diurnal of SecVOC1 shows a rapid enhancement starting from around 7:00-8:00 LT and declines continuously after 13:00 LT. This indicates that while many of these ions can be formed rapidly during daytime, they may have a short lifetime owing to partitioning to the condensed phase and/or heterogeneous processes.

Line 287: "Major fractions of alkyl nitrates (RONO2) are detected as (ROH+) fragment ions by the PTR-ToF-MS" - please cite Aoki et al: "Detection of C1–C5 alkyl nitrates by proton transfer reaction time-of-flight mass spectrometry" or similar

**Reply: We thank the referee for the suggestion. We added the reference. (Now Line 319)**

Line 350: "But more importantly, differences in SecVOC \*is probably owing to that chemical ...\*" - please rephrase; hard to follow

**Reply: We revised the sentence.**

Line 388-393. More importantly, the difference in SecVOC is probably due to different oxidation conditions at the two sites. Owing to the high mixing ratio of  $NO_x$  and the suppression of oxidants like OH radicals, the chemical oxidation of primary VOCs occurs to a greater extent downwind of urban emission sources. As shown in Fig. S8, high mixing ratios of SecVOC at MRIU originate from northwest directions, different from the origins of local primary factors (southeast). Besides, the longer oxidation time may be another reason of the higher SecVOC mixing ratios at MRIU.

**Reference**

- Bruns, E. A., Slowik, J. G., El Haddad, I., Kilic, D., Klein, F., Dommen, J., Temime-Roussel, B., Marchand, N., Baltensperger, U., and Prévôt, A. S. H.: Characterization of gas-phase organics using proton transfer reaction time-of-flight mass spectrometry: fresh and aged residential wood combustion emissions, Atmos. Chem. Phys., 17, 705-720, 10.5194/acp-17-705-2017, 2017.
- Gentner, D. R., Worton, D. R., Isaacman, G., Davis, L. C., Dallmann, T. R., Wood, E. C., Herndon, S. C., Goldstein, A. H., and Harley, R. A.: Chemical Composition of Gas-Phase Organic Carbon Emissions from Motor Vehicles and Implications for Ozone Production, Environmental Science & Technology, 47, 11837-11848, 10.1021/es401470e, 2013.
- McDonald, B. C., de Gouw, J. A., Gilman, J. B., Jathar, S. H., Akherati, A., Cappa, C. D., Jimenez, J. L., Lee-Taylor, J., Hayes, P. L., McKeen, S. A., Cui, Y. Y., Kim, S.-W., Gentner, D. R., Isaacman-VanWertz, G., Goldstein, A. H., Harley, R. A., Frost, G. J., Roberts, J. M., Ryerson, T. B., and Trainer, M.: Volatile chemical products emerging as largest petrochemical source of urban organic emissions, Science, 359, 760-764, 10.1126/science.aaq0524, 2018.
- Nidhi Tripathi, L. K. Sahu, Liwei Wang, Purushotam Kumar, Pawan Vats, R. V. Satish, Deepika Bhattu, Ravi Sahu, Kashyap Patel, Neeraj Rastogi, Shashi Tiwari6, Dilip Ganguly, André S. H. Prévôt, and Sachchida N. Tripathi. Emission sources and photochemical processes of VOCs at urban and suburban sites of New Delhi, India, in preparation
- Sekimoto, K., Koss, A. R., Gilman, J. B., Selimovic, V., Coggon, M. M., Zarzana, K. J., Yuan, B., Lerner, B. M., Brown, S. S., Warneke, C., Yokelson, R. J., Roberts, J. M., and de Gouw, J.: High- and low-temperature pyrolysis profiles describe volatile organic compound emissions from western US wildfire fuels, Atmos. Chem. Phys., 18, 9263-9281, 10.5194/acp-18-9263-2018, 2018.
- Platt, S. M., El Haddad, I., Pieber, S. M., Zardini, A. A., Suarez-Bertoa, R., Clairotte, M., Daellenbach, K. R., Huang, R. J., Slowik, J. G., Hellebust, S., Temime-Roussel, B., Marchand, N., de Gouw, J., Jimenez, J. L., Hayes, P. L., Robinson, A. L., Baltensperger, U., Astorga, C., and Prévôt, A. S. H.: Gasoline cars produce more carbonaceous particulate matter than modern filter-equipped diesel cars, Scientific Reports, 7, 4926, 10.1038/s41598-017-03714-9, 2017.
- Wu, R., and Xie, S.: Spatial Distribution of Ozone Formation in China Derived from Emissions of Speciated Volatile Organic Compounds, Environmental Science & Technology, 51, 2574-2583, 10.1021/acs.est.6b03634, 2017.
- Wu, R., and Xie, S.: Spatial Distribution of Secondary Organic Aerosol Formation Potential in China Derived from Speciated Anthropogenic Volatile Organic Compound Emissions, Environmental Science & Technology, 52, 8146-8156, 10.1021/acs.est.8b01269, 2018.